# Oscillatory rheotaxis of artificial swimmers in microchannels

Ranabir Dey [1,2✉], Carola M. Buness [1,3], Babak Vajdi Hokmabad [1,3], Chenyu Jin [1,4] & Corinna C. Maass [1,3,5✉]

Biological microswimmers navigate upstream of an external flow with trajectories ranging from linear to spiralling and oscillatory. Such a rheotactic response primarily stems from the hydrodynamic interactions triggered by the complex shapes of the microswimmers, such as flagellar chirality. We show here that a self-propelling droplet exhibits oscillatory rheotaxis in a microchannel, despite its simple spherical geometry. Such behaviour has been previously unobserved in artificial swimmers. Comparing our experiments to a purely hydrodynamic theory model, we demonstrate that the oscillatory rheotaxis of the droplet is primarily governed by both the shear flow characteristics and the interaction of the finite-sized microswimmer with all four microchannel walls. The dynamics can be controlled by varying the external flow strength, even leading to the rheotactic trapping of the oscillating droplet. Our results provide a realistic understanding of the behaviour of active particles navigating in confined microflows relevant in many biotechnology applications.

---

[1] Dynamics of Complex Fluids, Max Planck Institute for Dynamics and Self-Organization, Am Faßberg 17, 37077 Göttingen, Germany. [2] Department of Mechanical and Aerospace Engineering, Indian Institute of Technology Hyderabad, Kandi, Sangareddy, Telengana 502285, India. [3] Institute for the Dynamics of Complex Systems, Georg August Universität, 37077 Göttingen, Germany. [4] Physics Department, University of Bayreuth, 95440 Bayreuth, Germany. [5] Physics of Fluids Group, Max Planck Center for Complex Fluid Dynamics, MESA+ Institute and J. M. Burgers Center for Fluid Dynamics, University of Twente, PO Box 217, 7500 AE Enschede, The Netherlands. ✉email: ranabir@mae.iith.ac.in; corinna.maass@ds.mpg.de

Microswimmers, both biological and their artificial analogues, need to respond to external flows: classically, the locomotion of organisms in shear flows by continuous reorientation in response to changes in external velocity gradients is called rheotaxis[1]. Positive rheotaxis, i.e. upstream navigation, has been found for microorganisms in confined environments ranging from biological, like sperm cells in the reproductive tract[2] and bacteria in the upper urinary tract[3], to medical, like E. coli in catheters[4].

Microorganisms, like bacteria, swim upstream, often in an oscillatory manner, by utilising a combination of mechanisms stemming from the complex shape and motion of the microorganisms, hydrodynamic interaction with the confining surfaces in the vicinity, and the characteristics of the ambient flow[5–13]. Specifically for flagellates, the interaction of the moving flagella with the shear flow plays a critical role in dictating the rheotactic trajectory[7,12,13]. For artificial microswimmers, e.g. Janus particles and active droplets, in the absence of asymmetric and/or counter-rotating components, like flagella, most of the aforementioned rheotactic mechanisms are inapplicable. Therefore, the rheotactic characteristics of artificial microswimmers should not be predicted a priori based on observations related to the rheotaxis of biological microswimmers.

In recent years, the scientific community has started addressing the immense potential of artificial microswimmers in applications like targeted cargo delivery[14], including drug delivery[15], and water remediation[16,17]. In the majority of these applications, the microswimmers are inevitably required to navigate external flows, i.e. to rheotax. Despite this fact, the quantitative understanding of rheotaxis in artificial microswimmers is surprisingly limited. Only recently it was demonstrated that spherical Janus particles near a surface exhibit robust cross-stream migration at a definite orientation relative to a uniform shear flow[18,19]. Furthermore, the rheotactic characteristics of gold-platinum Janus rods was shown to depend on the interfacial gold/platinum length ratio[20] and the shear flow strength[21]. Efforts were also made to control the rheotactic characteristics of Janus rods using surface acoustic waves[22]. Rheotactic influences were also investigated and numerically modelled for magnetically actuated helical swimmer models[23,24].

Commonly, experiments on rheotaxing artificial microswimmers focus on geometries near a single surface and not a strong confinement where simultaneous interaction with multiple surfaces in presence of a non-uniform shear flow is possible. Furthermore, the present understanding is primarily based on the behaviour of Janus particles, and data is lacking for other artificial microswimmer platforms. Lastly, the interaction of external flow with the distribution of fuel and waste products in the vicinity or 'chemical trail' of an artificial microswimmer, which could influence the swimming dynamics, has not been explored yet.

In this study, we investigate the rheotaxis of finite-sized active droplets in a non-uniform, pressure-driven shear flow in the strong confinement of a microchannel.

Self-propelling active droplets are essentially oil/aqueous droplets in aqueous/oil-based surfactant media which are intrinsically symmetric, unlike Janus particles[25,26]. Therefore, oscillatory dynamics in quasi-2D, where these droplets exhibit a ballistic motion under no-flow conditions, can be unequivocally attributed to rheotactic effects. Moreover, they are simpler to generate and manipulate, and provide a promising and versatile platform for many of the envisioned applications for artificial microswimmers[14,15]. Briefly, the droplet activity is mediated by the following mechanism: The droplet solubilizes gradually into a micellar nanoemulsion. Starting from a dynamically unstable, radially symmetric solubilization state, the interfacial surfactant density spontaneously develops self-sustaining gradients, leading

the droplet to self-propel via the associated Marangoni stresses[27,28].

In the experiments under imposed flow shown here, the isotropic, active droplet swims upstream in a steady oscillatory trajectory despite its spherically symmetric shape, previously unobserved for isotropic microswimmers. Between the channel boundaries, it periodically accelerates, decelerates, and continuously changes its swimming orientation. During such oscillatory positive rheotaxis, the chemical (filled micelle) trail of the active droplet also traces a periodic pattern whose orientation closely follows the swimming orientation.

General features of oscillatory upstream rheotaxis and swinging downstream drift, for weak and strong counterflows respectively, have been previously traced back analytically to the interaction of a generic force dipole with a Poiseuille type shear flow[29]. However, such idealised, point-sized, strong pusher/puller assumptions might not be applicable to finite sized squirmers in strong confinements. In our analysis, we use a changed set of singularities accounting for the finite size of the droplet and its hydrodynamic interaction with all four walls of the microchannel; which reproduces essential non-harmonic features of the experimentally observed oscillatory rheotaxis, particularly the trajectory morphology and changes in orientation and acceleration dynamics. These dynamical changes will in real-life applications significantly affect the behaviour of microswimmers in narrow confinements.

While the rheotaxis of active droplets in confinements in itself deserves a detailed study, narrow channels are of particular interest, since this geometry applies to almost all conceivable problems in nature or technological application. We believe that our study of this oscillatory rheotactic behaviour in strong confinement will aid in conceptualising and designing both in vitro and in vivo applications, like targeted cargo delivery, which involve navigation of microswimmers in confinements, specifically their dynamics near boundaries.

## Results

**Oscillatory rheotaxis of an active droplet**. In this work, the active droplets consist of CB15 oil (dia. $2R_d \approx 50\,\mu m$) in supramicellar aqueous solutions of the ionic surfactant TTAB at 7.5 wt.%. We observe the swimming behaviour of these self-propelling droplets in a narrow microchannel (width $2w \approx 100\,\mu m$; height $2h \approx 2R_d \approx 52\,\mu m$) under an external pressure-driven flow using bright-field microscopy (Fig. 1a). The microchannel is fabricated from PDMS using photolithography and soft lithography techniques, while the pressure-driven flow is actuated and maintained using a syringe pump. The average imposed flow speed, $u_f^a \approx |\int_{-w}^{w} \langle u_x(x, y, z = 0) \rangle_x \, dy|/2w$, (Fig. 1b) is varied in the range of $u_f^a \approx 7.7 - 18.1\,\mu m s^{-1}$, corresponding to an imposed flow rate per channel in the range $\approx 40 - 94\,pL\,s^{-1}$. We define the coordinate system such that the origin is at the microchannel centerline, and the $x$ axis points upstream (Fig. 1b). $u_f^a$ is evaluated using PIV analysis, for which the aqueous surfactant solution is seeded with tracer particles. The order of magnitude of the resulting shear is $O(\dot{\gamma}) \sim u_f^a/w \approx 0.15 - 0.36\,s^{-1}$.

The droplet trajectory and *rheotactic* swimming speed in the external flow, $v_r \equiv |\mathbf{v}_r|$, given by the velocity vector $\mathbf{v}_r$ tracked in the laboratory frame, are extracted from the bright-field microvideographs using in-house Python code (see the Methods section for the experimental and post-processing details). We also define an intrinsic velocity $\mathbf{v}_0$ and orientation $\hat{\mathbf{e}} = \mathbf{v}_0/|\mathbf{v}_0|$ of the swimming droplet, where rheotactic speed and orientation have been corrected for the translational and rotational effects of the ambient velocity field at the instantaneous position of the droplet (see the Methods section). In the subsequent discussions, we non-

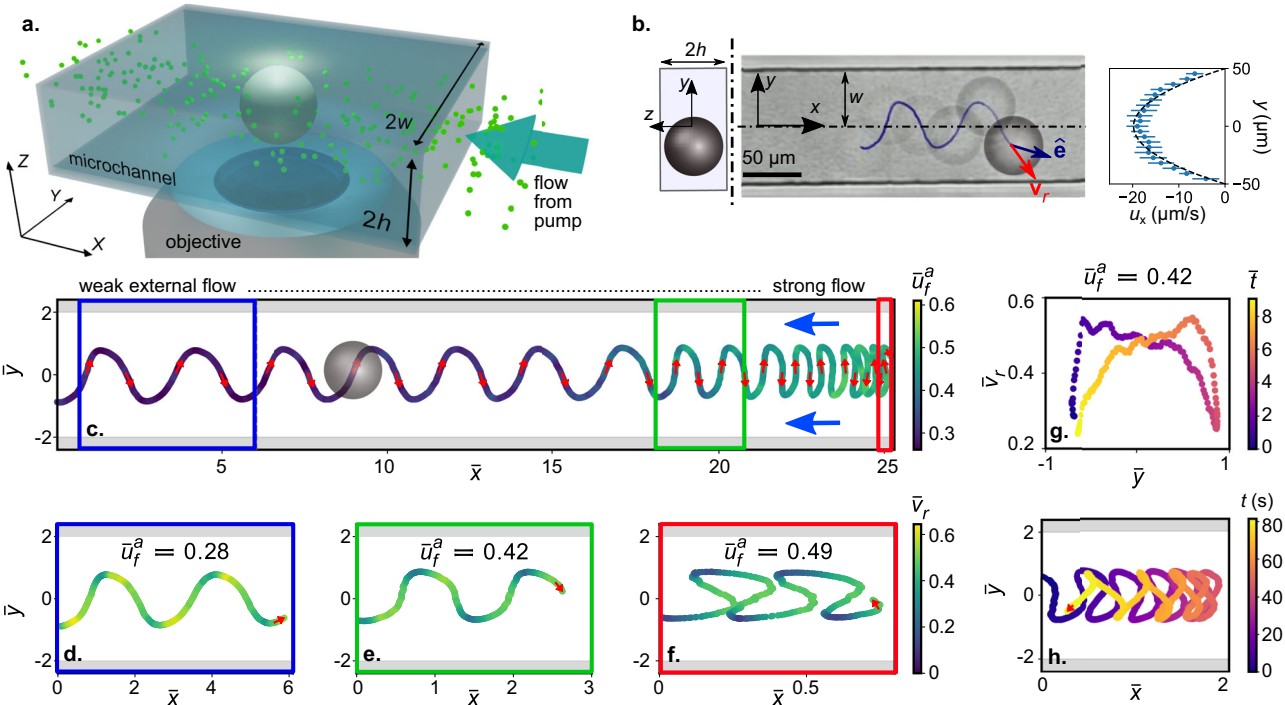

**Fig. 1 Oscillatory upstream (positive) rheotaxis of an active droplet in a microchannel for increasing strength of an imposed pressure-driven flow.**
**a** Microscopy setup schematic showing the active droplet in a narrow microchannel with height $\approx 52\,\mu m$ (droplet diameter $2R_d \approx 50\,\mu m$). **b** Time-lapse superposition from bright-field microscopy, showing the active droplet oscillating upstream, with a blue line tracking the droplet centroid. Right side: PIV-measured velocity profile in the plane of oscillation for the imposed pressure-driven flow, with a quadratic fit (dashed). Data points and error bars refer to average values and standard deviation of profiles measured along $x$. **c** The oscillatory rheotactic trajectory of the active droplet under increasing imposed flow, colour coded by the non-dimensional average flow velocity $\bar{u}_f^a$. **d-f** Variation in the non-dimensional rheotactic velocity $\bar{v}_r$ over the oscillatory trajectory for increasing values of $\bar{u}_f^a$; coloured boxes correspond to data ranges in (**c**). **g** Variation in $\bar{v}_r$ with the transverse location of the active droplet over one wavelength of the oscillation for $\bar{u}_f^a = 0.42$, colour coded by the non-dimensional time, $\bar{t}$. **h** Trapping of the oscillating active droplet within a region comparable to its diameter for over a minute, by adjusting $\bar{u}_f^a$.

dimensionalize spatial variables by $R_d$ (particularly, $\bar{h} \approx 1$), velocities by the intrinsic swimming speed $v_0 \equiv |\mathbf{v}_0| \approx 29.5 \pm 2\,\mu m$ $s^{-1}$, and times by $R_d/v_0$. All such non-dimensional quantities are denoted by overbars.

We made the following experimental observations. In the absence of the external flow, an active droplet self-propels in the microchannel following a linear trajectory while adhering to one of the side walls (see Supplemental Video S1). On actuating the pressure-driven flow in the direction opposite to the droplet motion, it swims upstream in a characteristic non-harmonic, channel-wide oscillatory trajectory in the X–Y plane (Fig. 1b, also see Supplemental Videos S2 and S3). With increasing $\bar{u}_f^a$, the active droplet exhibits persistent upstream oscillation with decreasing wavelength of the oscillatory trajectory (Fig. 1c, also see Supplemental Fig. 1). For a fixed value of $\bar{u}_f^a$, $\bar{v}_r$ varies along the trajectory in a periodic manner (Fig. 1d–f). Specifically, $\bar{v}_r$ sharply reduces as the droplet approaches a side wall (Fig. 1g). However, in the immediate vicinity of the side wall, the droplet accelerates, and $\bar{v}_r$ reaches its maximum value. Thereafter, $\bar{v}_r$ gradually reduces as the droplet orients away and swims towards the opposite side wall while advancing upstream. The aforementioned variation in $\bar{v}_r$ is repeated as the droplet crosses the channel centre-line and approaches the opposite side wall.

Beyond a threshold value of $\bar{u}_f^a \approx 0.53$ (cf. colour code in Figs. 1c and 2a), the droplet eventually cannot translate upstream in the laboratory frame. Interestingly, it is therefore possible to trap the oscillating droplet within a small region by judiciously tuning $\bar{u}_f^a$ about this threshold (Fig. 1h, also see Supplemental Video S2). Finally, above $\bar{u}_f^a \approx 0.53$, the active droplet drifts

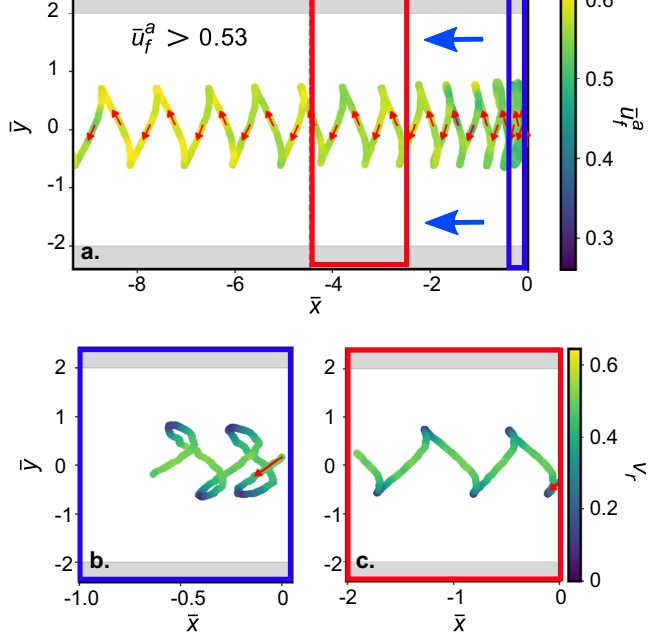

**Fig. 2 Downstream drift of the active droplet for strong counterflow.**
**a** Swinging trajectory of the active droplet as it drifts downstream, with increasing $\bar{u}_f^a$ above a threshold value ($\approx 0.53$), colour coded by $\bar{u}_f^a$, blue arrows marking the imposed flow direction. **b, c** Variation in $\bar{v}_r$ over the swinging trajectory for increasing values of $\bar{u}_f^a$; coloured boxes correspond to data ranges in (**a**).

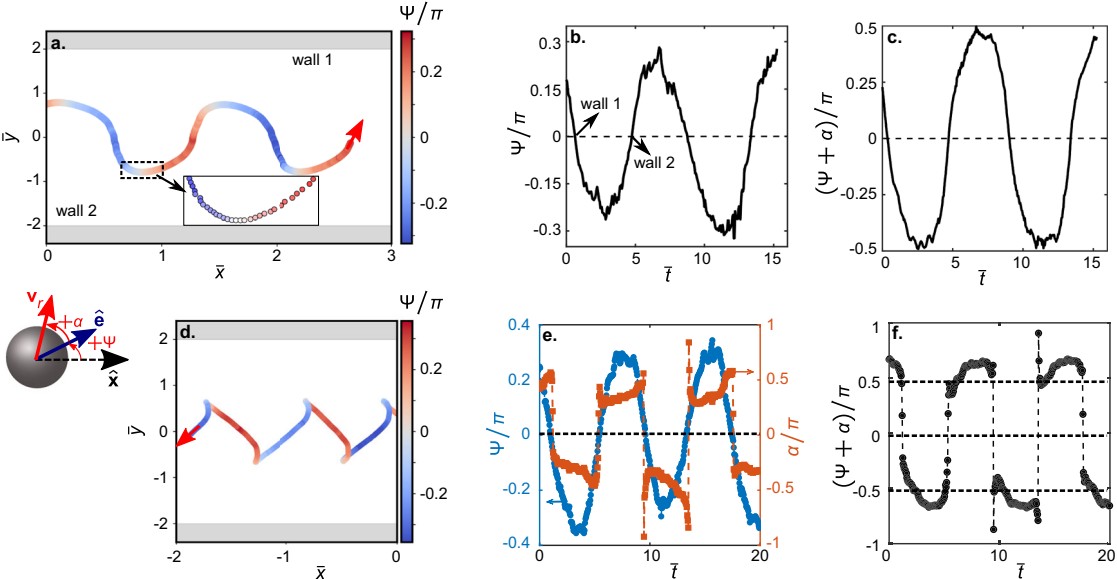

**Fig. 3 Intrinsic and rheotactic droplet orientations during upstream rheotaxis and downstream drift. a** Variation in the intrinsic orientation angle $\Psi$ of the active droplet over the oscillatory rheotactic trajectory for $\bar{u}_f^a = 0.42$. $\Psi$ is defined as the angle between droplet orientation $\hat{\mathbf{e}}$ and horizontal $\hat{\mathbf{x}}$ unit vectors (see schematic), with positive $\Psi$ representing anti-clockwise orientation about $+\hat{\mathbf{x}}$. The uncertainty in the measurement of $\Psi$ is $\approx$11%. **b** Periodic variation in $\Psi$ with the non-dimensional time ($\bar{t}$) during upstream rheotaxis. **c** Variation in the orientation of the rheotactic/tracked velocity vector $\mathbf{v}_r$ (see schematic) relative to $+\hat{\mathbf{x}}$, as given by ($\alpha + \Psi$), with $\bar{t}$. **d** Variation in $\Psi$ over the swinging trajectory during downstream drift of the active droplet. **e** Variations in $\Psi$ and $\alpha$, and (**f**) variation in ($\alpha + \Psi$) with $\bar{t}$ during the downstream drift. Note that the active droplet is always oriented upstream ($|\Psi|/\pi < 0.5$), even as it drifts downstream.

downstream in a swinging trajectory (Fig. 2, also see Supplemental Video S2).

For clarification, we note that we and others have previously found oscillatory motion even in a fully quiescent medium using CB15's nematic isomer 5CB as the oil phase[30,31]. However, in that case the oscillatory instability is caused by a nematoelastic torque which is absent in isotropic oils like 5CB.

We note three characteristic features of the downstream drift of the active droplet. First, the active droplet drifts downstream for $\bar{u}_f^a \gtrsim 0.53$, significantly below $\bar{u}_f^a \gtrsim 1$, which one can intuitively think of as an upper boundary to the threshold between upstream and downstream oscillation. Second, the wavelength of the swinging drift trajectory increases with increasing $\bar{u}_f^a$ (Fig. 2a, also see Supplemental Fig. S1), which is opposite to the trend observed during the oscillatory upstream rheotaxis (Fig. 1c). Finally, the periodic variation in $\bar{v}_r$ over the swinging downstream trajectory remains qualitatively similar to that observed during upstream oscillation (Fig. 2b, c).

**Variations in the intrinsic and rheotactic (translational) orientations of the active droplet**. To further quantify the oscillatory rheotactic characteristics, we extract the variation in the intrinsic orientation $\hat{\mathbf{e}}$ of the active droplet (see the Methods section). This is shown here by the variation in the angle $\Psi$ which defines the orientation of the unit vector $\hat{\mathbf{e}}$ relative to $+\hat{\mathbf{x}}$ in the X–Y plane via $\hat{\mathbf{e}} = \cos\Psi\hat{\mathbf{x}} + \sin\Psi\hat{\mathbf{y}}$ (see schematic in Fig. 3). Here, $\Psi \in \{-\pi, \pi\}$, and $\Psi$ is positive when measured in an anti-clockwise sense from $+\hat{\mathbf{x}}$. $\Psi$ varies in a periodic manner over the oscillatory rheotactic trajectory (Fig. 3a). As the active droplet approaches a side wall, it continuously orients itself to become parallel to that wall ($|\Psi|$ reduces; Fig. 3a, b). Eventually, the droplet microswimmer becomes parallel to the wall ($\Psi = 0$) in its immediate vicinity for a brief period (residence time $\approx$ 0.17 s at $|\Psi| < 0.1\pi$; Fig. 3a inset, b). Subsequently, the droplet orients away from the adjacent wall and towards the opposite side wall ($|\Psi|$

increases; Fig. 3a, b). The droplet eventually crosses the channel centre-line with the maximum cross-stream orientation (Fig. 3b), and begins to approach the other wall with decreasing $|\Psi|$ as before (Fig. 3b).

Note that the rheotactic direction of the active droplet is different from its intrinsic orientation. We account for it by additionally defining an offset angle $\alpha$ between $\hat{\mathbf{e}}$ and $\mathbf{v}_r$ (schematic in Fig. 3; also see Supplemental Fig. 2). The variation in the rheotactic direction relative to $+\hat{\mathbf{x}}$ is then given by the angle $\Psi + \alpha$ (Fig. 3c). $\Psi + \alpha$ varies smoothly about 0 in the vicinity of the side walls (Fig. 3c). This implies that the lab-frame orientation of the active droplet becomes truly parallel to the side walls in their immediate vicinity, and the droplet does not crash into the side walls during the oscillatory upstream rheotaxis.

We also evaluate the variations in the intrinsic and translational orientations of the active droplet during the swinging downstream drift (Fig. 3d–f). Notably, the droplet microswimmer always remains oriented upstream ($|\Psi|/\pi < 0.5$) even as it moves downstream (Fig. 3d, e). However, during the downstream drift, the resulting velocity field makes $\mathbf{v}_r$ orient downstream ($|\Psi + \alpha|/\pi > 0.5$) as the droplet traverses the width of the channel (Fig. 3f). While $\Psi$ varies smoothly even during the downstream motion, $\alpha$ undergoes sharp variations near the side walls (Fig. 3e). The latter is reflected in the variation of $\Psi + \alpha$ (Fig. 3f), and facilitates the downstream motion of the active droplet. In summary, the active droplet does not self-propel downstream, but it is pushed there in a swinging trajectory for $\bar{u}_f^a > 0.53$.

**Orientation of the chemical trail of the active droplet during oscillatory rheotaxis**. From our previous study on droplet motility in a quiescent medium we know that the chemical (filled micelle) field generated by the active droplet can locally interact with it, altering the swimming orientation[28]. We should therefore estimate whether similar secondary chemical interactions play a role in the rheotactic dynamics described above. To this end, we visualise the trail of filled micelles in the wake of the active

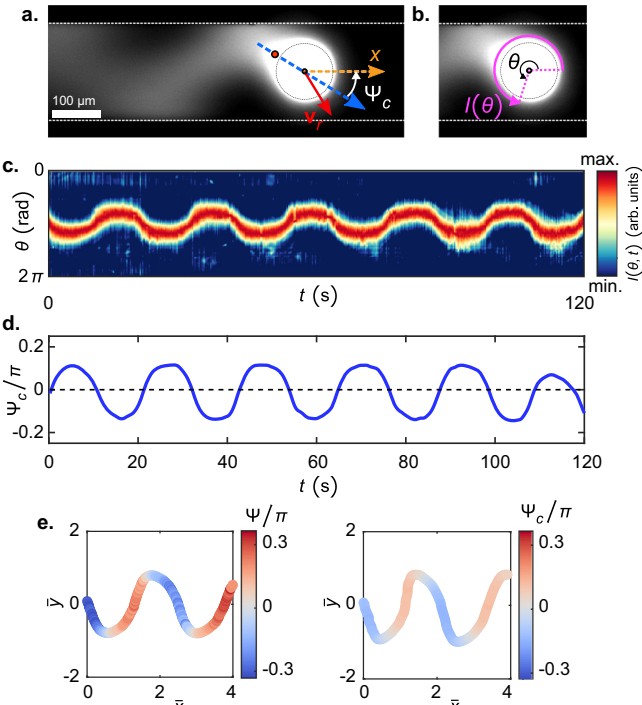

**Fig. 4 Characteristics of the chemical (filled micelle) trail of the active droplet during oscillatory rheotaxis. a** A fluorescence microscopy image showing the filled micelle trail of the active droplet during the oscillatory rheotaxis. The trail is visualised by doping the droplet with Nile red dye, which co-moves with the oil into filled micelles. **b** Definition of the polar angle $\theta$, taken counter-clockwise from $\hat{x}$, and the intensity profile $I(\theta)$ around droplet interface. **c** Kymograph of $I(\theta)$. **d** Temporal variation of the 'chemical angle' $\Psi_c$ which is supplementary to the angle $\theta(I_{max.})$ of maximum $I$ (see (a)). **e** Comparison between the intrinsic orientation of the active droplet, $\Psi$, and $\Psi_c$ over the oscillatory rheotactic trajectory for $\bar{u}_f^a \approx 0.42$. The chemical trail is closely aligned with the swimming orientation, and is not significantly distorted by the resulting flow field.

droplet during the oscillatory upstream rheotaxis using a fluorescent dye (Fig. 4a) which co-migrates with the oil molecules into the filled micelles (see Methods section for details). This endeavour also provides general insight into the behaviour of the chemical trail during the rheotaxis of artificial microswimmers; an aspect which has not been investigated before.

We extract the fluorescence intensity profile $I(\theta)$ emitted by the filled micelle field around the droplet interface over time (Fig. 4b), and map it onto a kymograph (Fig. 4c). We find that the orientation of the filled micelle trail of the droplet also follows a periodic variation during the oscillatory upstream rheotaxis. To quantify this periodic orientational variation of the filled micelle trail, we plot the 'chemical angle' $\Psi_c$ during the oscillatory rheotaxis in Fig. 4d. $\Psi_c$ is supplementary to the extracted orientation angle of $[I(\theta)]_{max}$ relative to $+\hat{x}$, i.e. $\Psi_c = \pi - \theta_{max}$. (Fig. 4a). Here, we assume that the location of $[I(\theta)]_{max}$ approximately coincides with the location of the maximum filled micelle concentration, which is at the posterior stagnation point (albeit in the droplet reference frame)[28]. Under quiescent conditions, the swimming orientation $\Psi$ and $\Psi_c$ are constant and identical, $\Psi = \Psi_c = $ const., as the orientation of the active droplet coincides with the anterior stagnation point. Therefore, for the present scenario, any offset between the variations in $\Psi$ and $\Psi_c$ will give an indirect measure of a distortion of the filled micelle trail orientation due to the ambient velocity field during rheotaxis within the microchannel, and such distortions can

trigger changes in the intrinsic swimming orientation[28]. However, a comparison between $\Psi$ and $\Psi_c$ over the rheotactic trajectory (Fig. 4e) shows that these are quite similar (also see Supplemental Fig. 3).

**Understanding the oscillatory rheotaxis based on hydrodynamic interaction.** From the trail analysis above we conclude that the filled micelle trail is not significantly distorted by the combination of active and imposed flows, and that it is safe to assume that the filled micelle field does not significantly interfere with the rheotactic dynamics via local chemical interactions. Therefore, in the following purely hydrodynamic model, we neglect the coupling between the chemical field and the hydrodynamics for the active droplet[32]. We assume that for an axisymmetric, finite-sized spherical microswimmer devoid of any chiral and/or counter-rotating components, the mechanism for rheotaxis will primarily stem from the interplay of the hydrodynamic interaction with the confining walls[33] and the change in swimming orientation due to the external shear flow (the Bretherton-Jeffery effect)[34]. We will confirm below by comparison to our experimental data whether these assumptions are sufficient to capture the characteristic oscillatory dynamics of the active droplet.

We therefore describe the pusher-type droplet microswimmer by an equivalent squirmer model comprising of a superposition of a positive force dipole ('fd'), a source dipole ('sd'), and a source quadrupole ('sq') (see the Methods section, as well as[32,33,35]). Accordingly, the velocity field of the swimming droplet can be written as:

$$\mathbf{u}_s = \alpha \mathbf{u}_{fd}(\mathbf{r} - \mathbf{r}_0; \hat{\mathbf{e}}(\Psi)) + \beta \mathbf{u}_{sd}(\mathbf{r} - \mathbf{r}_0; \hat{\mathbf{e}}(\Psi)) + \gamma \mathbf{u}_{sq}(\mathbf{r} - \mathbf{r}_0; \hat{\mathbf{e}}(\Psi)) \tag{1}$$

Here, $\alpha$, $\beta$, and $\gamma$ are the non-dimensional strengths of the force dipole, source dipole, and the source quadrupole singularities, $\mathbf{r}$ is the position vector, and $\mathbf{r}_0 = \bar{x}_0 \hat{x} + \bar{y}_0 \hat{y}$ is the location of the squirmer centroid. Subsequently, we use the method of images[29,32,33,35–38] to evaluate the velocity field ($\mathbf{u}_{HI}$) resulting from the hydrodynamic interaction of the squirmer with all four 'no-slip' walls of the microchannel, i.e. the two side walls normal to the X–Y plane, and the top and bottom walls parallel to the X–Y plane (see the Methods section for details). The total external velocity field $\mathbf{u}_t$ can be then written as:

$$\mathbf{u}_t = \mathbf{u}_s + \mathbf{u}_{HI} - \mathbf{u}_f$$

with

$$\mathbf{u}_{HI} = \alpha \sum \mathbf{u}_{fd}^*(\mathbf{r} - \mathbf{r}_0^*; \hat{\mathbf{e}}(\Psi)) + \beta \sum \mathbf{u}_{sd}^*(\mathbf{r} - \mathbf{r}_0^*; \hat{\mathbf{e}}(\Psi)) + \gamma \sum \mathbf{u}_{sq}^*(\mathbf{r} - \mathbf{r}_0^*; \hat{\mathbf{e}}(\Psi))$$

$$\mathbf{u}_f(\bar{y}, \bar{z}) = u_x(\bar{y}, \bar{z})\hat{x} = 6\bar{u}_f^a \left[1 - \left(\frac{\bar{y}}{\bar{w}}\right)^2\right]\hat{x} \tag{2}$$

$$+ 24\bar{u}_f^a \sum_{n=0}^{\infty} \frac{8(-1)^{n+1}}{(2n+1)^3 \pi^3} \text{sech}\left((2n+1)\frac{\pi}{4}\right) \times$$

$$\cos\left((2n+1)\frac{\pi}{2}\frac{\bar{y}}{\bar{w}}\right) \cosh\left((2n+1)\frac{\pi}{2}\frac{\bar{z}}{\bar{w}}\right)\hat{x}$$

Here, $\mathbf{u}_{fd/sd/sq}^*$ represents the velocity field due to the image system at a wall corresponding to a particular singularity, $\mathbf{r}_0^*$ is the location of the image singularities for a particular wall, $\sum$ represents the vector sum of $\mathbf{u}_{fd/sd/sq}^*$ for all the walls, and $\mathbf{u}_f$ is the imposed pressure-driven velocity field for a microchannel with a rectangular cross-section. Furthermore, there are two additional approximations involved here. First, Eq. (2) is a far-field approximation, and hence ignores that the droplet radius and

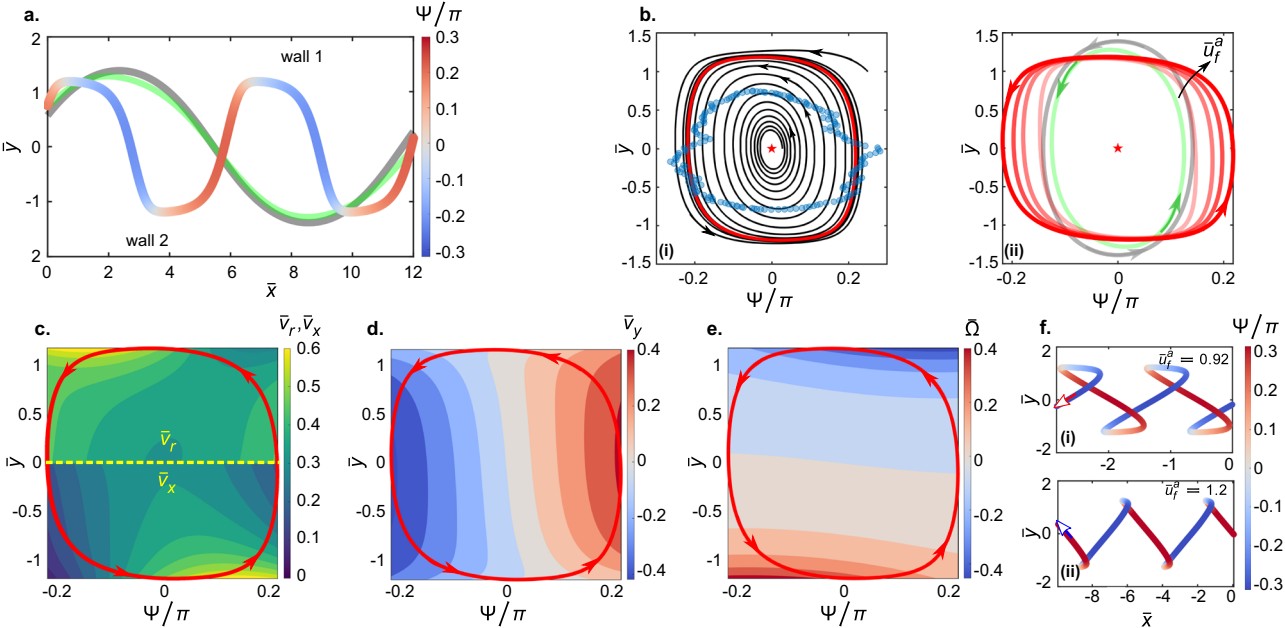

**Fig. 5 Dynamics of the oscillatory rheotaxis of the active droplet predicted by a hydrodynamic model. a** Variation in the swimming orientation ($\Psi$) of the active droplet over the oscillatory trajectory for $\bar{u}_f^a = 0.42$ evaluated using Eqs. (3)-(5). Grey and green trajectories represent predictions from two reduced order models for $\bar{u}_f^a = 0.42$: without any wall-microswimmer hydrodynamic interaction (grey), and without the finite-size effect ($\alpha = 0.16, \beta = 0, \gamma = 0$) (green). **b** (i) Trajectories in the $\Psi$-$\bar{y}$ phase space for the upstream rheotaxis of the active droplet at $\bar{u}_f^a = 0.42$. The microswimmer is initialised with a non-zero upstream orientation from the channel centre as well as from near the microchannel wall. In both the cases the microswimmer approaches the same closed trajectory (a stable limit cycle; solid red line). The blue markers represent the experimentally obtained stable trajectory. **b** (ii) Variation of the stable limit cycle with $\bar{u}_f^a$. Increasing opacity of the red line represents increasing $\bar{u}_f^a$, with the solid red line representing $\bar{u}_f^a = 0.42$ (as in (i)). Solid grey and green lines represent the stable limit cycles for the corresponding reduced order models for $\bar{u}_f^a = 0.42$ plotted in (**a**). **c** The stable trajectory of the droplet microswimmer in the $\Psi$-$\bar{y}$ phase space for $\bar{u}_f^a = 0.42$ against the contour plots representing $\bar{v}_r$ (top half) and the non-dimensional x-component of the rheotactic velocity ($\bar{v}_x$) (bottom half). **d, e** The stable trajectory of the droplet microswimmer against the contour plot representing the non-dimensional y-component of the rheotactic velocity ($\bar{v}_y$) and the non-dimensional angular velocity of the microswimmer ($\bar{\Omega}$) respectively. **f** (i-ii) Trajectories and variations in orientation during the downstream drift for increasing values of $\bar{u}_f^a$ evaluated using Eqs. (3)–(5).

wall distance are of similar order. Second, $\mathbf{u}_t$ is approximate to the leading order as we consider a total of twelve image systems - images for force dipole, source dipole, and source quadrupole singularities at each of the four walls, instead of the infinite series of images that should be considered to satisfy the no-slip boundary condition at the walls[36,37].

We use Faxén's first law for a force-free sphere[33,35,39] to evaluate the rheotactic velocity of the squirmer in response to $\mathbf{u}_t$ (see the Methods section). Accordingly, the $x$ and $y$ components of the rheotactic (translational) velocity can be evaluated as:

$$
\begin{aligned}
\bar{v}_x = \cos\Psi &- 6\bar{u}_f^a\left[1 - \left(\frac{\bar{y}_0}{\bar{w}}\right)^2 - \frac{1}{3\bar{w}^2}\right] \\
&+ 24\bar{u}_f^a\sum_{n=0}^{\infty}\frac{8(-1)^n}{(2n+1)^3\pi^3}\frac{\cos\left((2n+1)\frac{\pi}{2}\frac{\bar{y}_0}{\bar{w}}\right)}{\cosh\left((2n+1)\frac{\pi}{4}\right)} \\
&+ \alpha\left[\pm\frac{0.875}{(\bar{y}_0\pm\bar{w})^2}\pm\frac{0.078125}{(\bar{y}_0\pm\bar{w})^4}\right]\sin 2\Psi \\
&+ \beta\left[\mp\frac{1}{4(\bar{y}_0\pm\bar{w})^3}\pm\frac{3}{16(\bar{y}_0\pm\bar{w})^5}-\frac{17}{16}\right]\cos\Psi \\
&+ \gamma\left[\mp\frac{3}{16(\bar{y}_0\pm\bar{w})^4}\mp\frac{5}{128(\bar{y}_0\pm\bar{w})^6}\right]\cos\Psi
\end{aligned} \quad (3)
$$

$$
\begin{aligned}
\bar{v}_y = \sin\Psi &+ \alpha\left[\mp\frac{3(1-3\sin^2\Psi)}{8(\bar{y}_0\pm\bar{w})^2}\pm\frac{(7-11\sin^2\Psi)}{64(\bar{y}_0\pm\bar{w})^4}\right] \\
&+ \beta\left[\mp\frac{1}{(\bar{y}_0\pm\bar{w})^3}\pm\frac{1}{8(\bar{y}_0\pm\bar{w})^5}-\frac{17}{16}\right]\sin\Psi \\
&+ \gamma\left[\pm\frac{3(3+7\sin^2\Psi)}{16(\bar{y}_0\pm\bar{w})^4}\pm\frac{5(11-7\sin^2\Psi)}{128(\bar{y}_0\pm\bar{w})^6}\right]
\end{aligned} \quad (4)
$$

Here, each of the terms with $\pm/\mp$ represent two terms due to the image systems at the two side walls; the first and second signs represent the contribution from the image systems at wall 2 and wall 1 respectively (Fig. 5a), while the image systems at the top and bottom walls solely contribute by a constant 17/16 in the source dipole term. Finally, the temporal change in the intrinsic swimming orientation of the squirmer is given by $\dot{\hat{\mathbf{e}}} = \bar{\Omega}\times\hat{\mathbf{e}}$, where $\bar{\Omega} = \frac{1}{2}\nabla\times(\mathbf{u}_{HI} - \mathbf{u}_f)$ is the angular velocity of the squirmer according to Faxén's second law for a torque-free sphere[33,35]. Thus, the rate of change of the intrinsic swimming orientation can be evaluated as:

$$
\begin{aligned}
\dot{\Psi} = -6\bar{u}_f^a\left(\frac{\bar{y}_0}{\bar{w}^2}\right) &+ 12\frac{\bar{u}_f^a}{\bar{w}}\sum_{n=0}^{\infty}\frac{4(-1)^n}{(2n+1)^2\pi^2}\frac{\sin\left((2n+1)\frac{\pi}{2}\frac{\bar{y}_0}{\bar{w}}\right)}{\cosh\left((2n+1)\frac{\pi}{4}\right)} \\
&\mp\alpha\frac{3}{16(\bar{y}_0\pm\bar{w})^3}\sin 2\Psi\pm\beta\frac{3}{8(\bar{y}_0\pm\bar{w})^4}\cos\Psi \\
&\pm\gamma\frac{3}{8(\bar{y}_0\pm\bar{w})^5}\sin 2\Psi
\end{aligned} \quad (5)
$$

Eqs. (4) and (5) form a system of coupled ordinary differential equations, which we solve numerically for $\bar{y}_0(\bar{t})$ and $\Psi(\bar{t})$.

Thereafter, Eq. (3) is numerically integrated to evaluate the corresponding $\bar{x}_0(\bar{t})$ (see the Methods section). Figure 5a shows the trajectory of the squirmer colour coded by $\Psi$ for $\bar{u}_f^a = 0.42$, as obtained from Eqs. (3)–(5). A comparison with Fig. 3a shows that the hydrodynamic model indeed matches the oscillatory rheotaxis of the active droplet for a finite-sized, weak pusher (cf.[40]) with $\alpha = 0.16$, $\beta = 0.26$, and $\gamma = -0.08$ (also see Supplemental Fig. 4 for a qualitative prediction about the rheotactic behaviour of puller-type microswimmers).

**Phase portraits.** We illustrate the orientational dynamics by plotting the rheotactic dynamics in the $\Psi$-$\bar{y}$ phase space. The steady upstream oscillation of the active droplet represents a stable limit cycle in the $\Psi$-$\bar{y}$ phase space (Fig. 5b(i); red line: theory; blue markers: experiment; also see Supplemental Fig. 5 for additional experimental limit cycles). Note that perfect upstream orientation at the microchannel centerline ($\Psi = 0, \bar{y} = 0$; star marker in Fig. 5b(i)) represents an unstable fixed point, as is generally the case for the phase space dynamics of pusher-type microswimmers[29,37]. With increasing $\bar{u}_f^a$, the active droplet oscillates upstream with sharper changes in $\Psi$ over $\bar{y}$ (red lines in Fig. 5b(ii)). This is due to the increase in $\bar{\Omega}$ with increasing vorticity of the background flow, since Faxén's second law sets a one-to-one relationship between the two quantities. Additionally, $\bar{v}_x$ reduces with increasing $\bar{u}_f^a$ (see Eq. (3)). The increase in $\bar{\Omega}$ and the simultaneous decrease in $\bar{v}_x$ combine to reduce the oscillation wavelength with increasing $\bar{u}_f^a$ (Fig. 1c). The finite size of the active droplet relative to the microchannel, as embodied by $\beta$, plays a pivotal role in the observed rheotactic dynamics. We demonstrate this by comparing the planar, as well as the phase space trajectories, with (colour coded/ solid red line) and without (green lines, $\beta = 0$) finite-size effects in Fig. 5a, b(ii). Due to its finite radius, the droplet has higher wall-induced $\bar{\Omega}$ resulting in enhanced variation in $\Psi$, and reduced $\bar{v}_r$ due to the influence of the top and bottom walls (see Eqs. (3) and (4)). For comparison, the grey lines in Fig. 5a, b(ii) represent the trajectory for an active particle in a pressure-driven flow without any hydrodynamic interaction, i.e. no contribution of any of the singularities and their image systems in Eqs. (3)–(5). Thus, while it is sufficient to use reduced order descriptions (grey and green trajectories), as in[29] to replicate oscillations in general (and hence stable limit cycles), those are quite distinct from the dynamics of an experimental finite-sized swimmer represented by the changed image system, as shown by our comparison to experimental data.

The variation of $\bar{v}_r$ and $|\Psi|$ along the oscillatory trajectory obtained using Eqs. (3)–(5) (top half of Fig. 5c) are also similar to the experimentally observed variations (Figs. 1e, g and 3a, b). At a constant value of $\bar{u}_f^a$, the active droplet begins to approach a side wall with a large $|\Psi|$ (oriented towards the wall). As the pusher-type droplet microswimmer approaches the side wall, it becomes increasingly difficult to swim by pushing liquid onto the impermeable wall while satisfying the incompressible nature of the generated flow field. This, along with the continuous reduction in $|\Psi|$ towards 0 due to the increasing $\bar{\Omega}$ (Fig. 5e), result in the stronger reduction in $|\bar{v}_y|$ (Fig. 5d) compared to the weaker increase in $\bar{v}_x$ (bottom half of Fig. 5c). Such wall-induced adjustment to the droplet swimming dynamics manifests in the reduction in $\bar{v}_r$ (Fig. 1g) with reducing $|\Psi|$ (Fig. 3b) during the approach to a microchannel side wall. In the immediate vicinity of a side wall, the droplet eventually becomes almost parallel to the wall ($|\Psi|$ varies about 0). Now, the droplet can swim efficiently by pushing liquid almost tangential to the wall. This manifests in the sharp increase in $\bar{v}_x$ (bottom half of Fig. 5c), with

smaller variation in $|\bar{v}_y|$ (Fig. 5d). Consequently, $\bar{v}_r$ increases sharply adjacent to a side wall (Fig. 1g).

Eventually, the droplet orients away from the side wall due to the strong $|\Omega|$. As the droplet moves away from the side wall, $|\Psi|$ increases due to the orientation of $\bar{\Omega}$ (Fig. 5e). This time, the wall-induced modifications result in the stronger reduction in $\bar{v}_x$ (bottom half of Fig. 5c) compared to the increase in $\bar{v}_y$ (Fig. 5d). Consequently, $\bar{v}_r$ again reduces (Fig. 1g) with increasing $|\Psi|$ (Fig. 3b). This dynamics repeats itself once the droplet microswimmer crosses the centerline of the microchannel, and approaches the opposite side wall resulting in the channel-wide oscillatory rheotaxis.

Eqs. (3)–(5) also describe the downstream drift of the active droplet (compare Fig. 5f and Fig. 2), corresponding to a similar phenomenon in point swimmers[29]. The hydrodynamic model recreates the two main features of the swinging downstream drift––the active droplet always remains oriented upstream, and the wavelength of the swinging trajectory increases with increasing $\bar{u}_f^a$ (Fig. 5f). The consistent inclusion of the hydrodynamic interaction of the finite-sized microswimmer with all the confining walls, while estimating $\bar{v}_r$ and $\dot{\Psi}$, can recreate the downstream drift for $\bar{u}_f^a < 1$. We also conclude that the theoretical model can be used to address the hydrodynamic trapping of the active droplet (Fig. 1h) by tuning $\bar{u}_f^a$ such that $\bar{\Omega}$ is strong and simultaneously $\bar{v}_x$ varies about 0. Finally, we note that the rheotactic dynamics of the active droplet is also dependent on the initial orientation of the droplet $\Psi_0$ relative to the imposed flow at the instant of actuation ($t = 0$). The oscillatory upstream rheotaxis, and its dependence on $\bar{u}_f^a$, as described above, are valid for an initially upstream oriented, or even weakly downstream oriented active droplet ($0 < |\Psi_0|/\pi < 0.55$, assuming $\mathbf{u}_f^a \propto -\hat{\mathbf{x}}$, (also see Supplemental Fig. 6). We show an experimental example for $\Psi_0 = \pi$ and $\bar{u}_f^a \approx 0.3$ in Supplementary Video S4, where the droplet remains at one side wall.

**Discussion**

In this study, we have investigated in experiment and theory a problem common in nature and artificial microswimmers, e.g. lab-on-a-chip applications: robust positive rheotaxis in microchannels under external flow. In our experiments we documented oscillatory upstream rheotaxis in microchannels for artificial swimmers consisting of active droplets. Although this type of oscillatory rheotaxis has been observed for bioswimmers like E. coli[12] and T. brucei[6], it has not been demonstrated unequivocally before for artificial microswimmers.

We model the oscillatory rheotaxis of the active droplets in the microchannel following a purely hydrodynamic approach of mapping the droplet on a finite-sized ideal squirmer, while accounting for its hydrodynamic interactions with the confining walls and the imposed shear flow. We achieve good quantitative agreement between experiment and theory even under the approximations of ignoring the chemical interactions and truncating the hydrodynamic images systems to the leading order for the walls of the microchannel.

Using phase portraits, we illustrate that this model features a robust limit cycle, as also demonstrated by the noticeable regularity and reproducibility of our recorded trajectories, distinct from limit cycles applicable to point swimmers.

From a general perspective, the key finding of this study is a comparative analysis between an experimental model system of maximal symmetry and its close theory representation to judge the efficacy of hydrodynamic interactions in explaining the rheotactic dynamics of microswimmers in confinements––an issue which has been deliberated upon so far based on solely

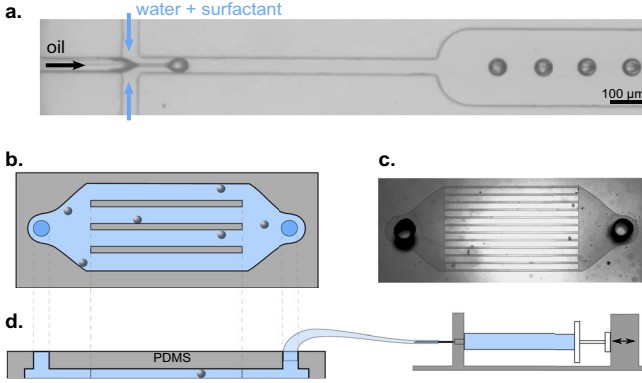

**Fig. 6 Microfluidics. a** micrograph of the flow focusing geometry for droplet generation. **b–d** Microchannel geometry setup: (**b**) Top view schematic of the used channel structure, **c** micrograph of the channel structure (channel width 100 μm), (**d**) side view schematic of the PDMS structure with connected syringe pump.

theoretical/numerical works[29,41–43]. It will be interesting to investigate in future studies whether the inclusion of near-wall hydrodynamics[44] and/or both phoretic and hydrodynamic interactions for the model microswimmer in the confinement, as done very recently[45], can bridge the remaining differences between experiment and our somewhat idealised model for the rheotaxis of artificial microswimmers––notably, the relative overestimation (underestimation) of the oscillation wavelength (angular velocity) during rheotaxis.

This in-depth investigation of the physical mechanisms underlying the rheotactic dynamics of artificial microswimmers will add valuable insight to conceptualising and designing practical micro-robotic applications. For one, we have demonstrated that robust hydrodynamic trapping is feasible via a simple feedback protocol for the imposed flow, providing a facile method to keep a motile microrobot in place. Furthermore, the effect a swimmer has on its environment will be determined by its behaviour in strong confinement and near interfaces like microfluidic walls, tubing or biological vessels. Here, our model provides a physically realistic picture of acceleration, orientation and residence dynamics (see Supplemental Fig. 7) mediated by hydrodynamic interactions with such interfaces.

## Methods

**Fabrication of micro-confinements**. We fabricated microfluidic chips in-house, using soft lithography techniques[46], as follows: CAD photomask transparencies were printed to our specifications at 128,000 dpi by JD Photo-Tools. We used the photomask to create a negative mold by UV exposure of a layer of the photoresist SU-8 3025 (MicroChem), which we had spin-coated onto a silicon wafer (Si-Mat).

We then poured a poly(dimethyl siloxane) (PDMS) mixture (SYLGARD 184 Silicone Elastomer Kit, DowCorning) with 10:1 weight ratio of base to cross-linker onto the mold and baked it for 2 h at 75 °C. We used two types of microfluidic geometries: a flow-focusing cross junction for the generation of monodisperse CB15 droplets (Fig. 6a), and multi-channel geometries with inlet and outlet reservoirs for the rheotactic experiments (Fig. 6b–d). For the droplet generation we used a glass slide as a coverslip for the channel, whereas for the microchannels used in the experiments we coated the glass with PDMS to have similar boundary conditions at all channel walls. To create an even PDMS surface, we placed the glass slide covered in liquid PDMS in a spin coater (Laurell, WS-650HZB-23NPPB) for 45 s at 500 rotations/min, and then baked it for 2 h at 75 °C. Finally, both channel structure and glass slide were treated with a partial pressure air plasma (Pico P100-8; Diener Electronic GmbH+Co. KG) for 30 s and then pressed together, bonding the two surfaces.

**Production of monodisperse CB15 droplets**. The walls of these microfluidic chips were chemically treated to hydrophilise the channels where aqueous surfactant solution flowed. We followed the recipe of Petit et al.[47]: First, the channel walls were oxidized by a 1:1 mixture of hydrogen peroxide solution (H$_2$O$_2$ at 30 wt%, Sigma-Aldrich) and hydrochloric acid (HCl at 37 wt%, Sigma-Aldrich). The mixture was flushed through the channels for 2 min. Then, the channel was rinsed

by flushing double distilled water for 30 s. Next, a 5 wt% solution of the positive polyelectrolyte poly(diallyldimethylammonium chloride) (PDADMAC, Sigma-Aldrich) was flushed for 2 min through the oxidized channel. The PDADMAC binds to the activated channel walls by ionic interactions. Finally, a 2 wt% solution of the negative polyelectrolyte poly(sodium 4-styrenesulfonate) (PSS, Sigma-Aldrich) was flushed for 2 min.

To generate droplets, we mounted syringes of oil and 0.1 wt% aqueous TTAB solution to a microprecision syringe pump (NEM-B101-02B; Cetoni GmbH), connected them to the two inlets of the microfluidic chip via Teflon tubing, and tuned the flow speed through the chip to reach the desired droplet diameter. Once the production was monodisperse and at a steady state, droplets were collected and kept in 0.1 wt% TTAB stock solution.

**Experimental protocol for visualising the rheotactic behaviour of active droplets**. To observe the activity of the droplets, we put them into a 7.5 wt% aqueous TTAB solution. To visualise the flow, and for subsequent PIV analysis, tracer particles (FluoSpheres$^{TM}$ carboxylate, 0.5 μm, red (580/605), Life Technologies Corporation) were added to the aqueous solution.

We observed the droplet motion in a PDMS chip with several parallel channels (Fig. 6), each of height $2h = 52$ μm and width of $2w = 100$ μm. The flow inside the channels was controlled by a syringe pump connected via Teflon tubing to the inlet of the channel structure. The outlet was left open and unconnected. For quantitative PIV, we used an inverted bright-field microscope (Olympus IX-81) at a magnification of 20× for observation, and recorded videos at a frame rate of 25 frames per second with a Canon (EOS 600D) DSLR camera (1920 × 1080 px). Given the numerical aperture $NA = 0.7$, magnification of 20×, and pixel size of the sensor 4.3 μm, the depth of focus can be estimated to ≈1.5 μm[48]. The objective was focused on the mid-plane of the channel, such that the PIV data was collected with good accuracy in in the plane of motion of the active droplet centroid.

Pixel-to-micrometre conversions were calibrated by imaging microstructures with known dimensions, including micrometre stages and channel structures.

To estimate the intrinsic speed $v_0$ of the swimmer, we placed ≈5 droplets from the same batch in a similar quiescent medium, i.e. a 7.5 wt% aqueous TTAB solution, in a rectangular reservoir 50 μm in height, recorded and analysed their motion for several minutes. We only analysed parts of the trajectories where the droplets were not interacting with any boundaries or each other.

**Post-processing of experimental data**. The droplet coordinates $(x_0, y_0)$ in each frame were extracted from video frames (bright-field microscopy images) using software written in-house in Python/openCV[49], via a sequence of constant background correction, threshold binarisation, blob detection by contour analysis, and minimum enclosing circle fits. Swimming trajectories were obtained from a frame-by-frame nearest-neighbour analysis[50]. The rheotactic droplet velocity components $(\dot{x}_0, \dot{y}_0)$ were calculated using the droplet position in each frame and the frame rate.

We analysed the flow in the aqueous swimming medium using the open source MATLAB package PIVlab[51], based on images extracted from videomicroscopy data with the open source software ffmpeg, and without any further preprocessing. Distances were calibrated in PIVlab by the known channel width of 100 μm. For the PIV analysis, we chose a region of interest across the channel cross-section in $y$ and of around 60 μm along the channel in $x$. For estimating the profile of the imposed flow in the plane of oscillation of the active droplet, we selected a region as far away from the droplet as possible, but for the evaluating the droplet orientation, we analysed the flow field directly upstream of the droplet. Within PIVlab, all settings were kept at their default values except for the interrogation area (Pass 1: 64, Pass 2: 32, Pass 3: 16, Pass 4: 8).

Further data analyses and plots were generated using Python's NumPy and matplotlib libraries. The rheotactic droplet velocity components $\dot{x}_0$, $\dot{y}_0$ can be written as

$$\dot{x}_0 = v_0 \cos \Psi + u'_x$$
$$\dot{y}_0 = v_0 \sin \Psi + u'_y. \tag{6}$$

This stems from the condition that $\mathbf{v}_r = \mathbf{v}_0 + \mathbf{u}'$. Here, $\mathbf{u}' = u'_x \hat{\mathbf{x}} + u'_y \hat{\mathbf{y}}$ is ideally the disturbance flow field at the centroid location of the droplet. $\mathbf{u}'$ was experimentally approximated from the PIV analysis in the immediate vicinity of the droplet, in the plane of oscillation. Note that $\dot{x}_0$ and $\dot{y}_0$ were also experimentally evaluated from post-processing of experimental data (see above). Hence, from the aforementioned equations the intrinsic droplet orientation $\Psi$ and the translational orientation (orientation of the rheotactic velocity vector) $\Psi + \alpha$ can be evaluated as

$$\Psi = \tan^{-1}\left(\frac{\dot{y}_0 - u'_y}{\dot{x}_0 - u'_x}\right) \text{ and } \Psi + \alpha = \tan^{-1}\left(\frac{\dot{y}_0}{\dot{x}_0}\right). \tag{7}$$

**Visualisation of the chemical (filled micelle) trail during rheotaxis**. We used fluorescent microscopy for direct imaging of the chemical trail secreted by the droplet[28,52]. We doped the oil phase with the fluorescent dye Nile red (Thermo Fisher Scientific). The dye molecules co-migrate with the oil phase into the filled micelles shed by the droplet. Experiments were observed on an Olympus IX73 microscope with a filter cube (Excitation filter ET560/40x, Beam splitter 585 LP and Emissions filter ET630/75m, all by Chroma Technologies). Images were

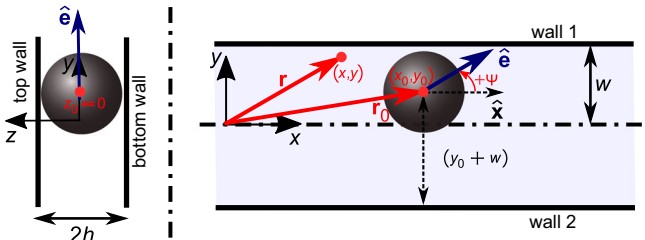

**Fig. 7 Parameters and variables.** Schematic of the active droplet in the microchannel with rectangular cross-section.

recorded using a 4 MP CMOS camera (FLIR Grasshopper3, GS3-U3-41C6M-C), at four frames per second and 4× magnification. Recorded images were analysed, using MATLAB, to extract the red light intensity profiles, $I(\theta)$, around the droplet close to the interface. We then identified the angular position of the maximum intensity point $[I(\theta)]_{max}$. The line from this point to the centre of the droplet determines the orientation of the chemical field and in turn yields the 'chemical angle' $\Psi_c$.

#### Details of the theoretical model

*Squirmer model.* We represent the flow field generated by the finite-sized droplet microswimmer by an equivalent squirmer model comprising of a force dipole singularity ('*fd*'; representing the effect of the net force-free, pusher-type swimming mechanism of the active droplet), a source dipole singularity ('*sd*'; representing the effect of the finite size of the spherical droplet microswimmer), and a source quadrupole singularity ('*sq*'; to capture the hydrodynamics closer to the micro-swimmer, i.e. in the near-field region)[32,33,35]. Accordingly, the velocity field of the swimming droplet can be written as in Eq. (1). Since, for the microchannel, the droplet is constrained to translate and orient only in the X–Y plane (Fig. 7), $\bar{z}_0$ and the orientation angle of the squirmer normal to the X–Y plane are always zero.

*Method of images.* We address the hydrodynamic interaction of the aforementioned squirmer with the confining walls of the microchannel using the method of images[29,32,33,35–38]. We consider the images of each of the three singularities for the four 'no-slip' walls of the microchannel, i.e. the two side walls normal to the X–Y plane (*wall 1* and *wall 2*), and the top and bottom walls parallel to the X–Y plane (Fig. 7). For reference, the vectorial forms of the velocity fields due to the image systems for the force dipole ($\mathbf{u}_{fd}^*(\mathbf{r} - \mathbf{r}_0^*; \hat{\mathbf{e}}(\Psi))$) and source dipole singularities ($\mathbf{u}_{sd}^*(\mathbf{r} - \mathbf{r}_0^*; \hat{\mathbf{e}}(\Psi))$) corresponding to *wall 2*, following[33], are shown in the Supplementary Material. For *wall 2*, the location of the image system corresponding to a singularity is $\mathbf{r}_0^* = \mathbf{r}_0 - 2(\bar{y}_0 + \bar{w})\hat{\mathbf{y}}$. Once the image of the source dipole singularity is evaluated, the image of the source quadrupole singularity, corresponding to *wall 2*, is subsequently computed as[33]

$$\left[ \mathbf{u}_{sq}^*\left(\mathbf{r} - \mathbf{r}_0^*; \hat{\mathbf{e}}(\Psi)\right) \right] = \left[ \nabla|_{\mathbf{r}_0} \mathbf{u}_{sd}^* \right][\hat{\mathbf{e}}] \qquad (8)$$

Note here that the $\nabla$ operator acts on the coordinates of the location of the squirmer centroid $(\bar{x}_0, \bar{y}_0, \bar{z}_0 = 0)$. The velocity fields associated with the images of the singularities corresponding to *wall 1* are evaluated in an identical manner considering that for this wall $\mathbf{r}_0^* = \mathbf{r}_0 - 2(\bar{y}_0 - \bar{w})\hat{\mathbf{y}}$. The velocity fields due to the image systems corresponding to the bottom and top walls of the microchannel are also computed following the same methodology as outlined in[33], considering the following details- (i) for the droplet microswimmer in the microchannel under consideration $\bar{z}_0 = 0$; (ii) the images are located at a distance $\sim R_d$ behind the walls; (iii) the orientation angle of the squirmer normal to the X–Y plane is zero. Once all the image systems have been evaluated, the total velocity field external to the squirmer can be written as in Eq. (2). The approximations involved here are explicitly mentioned in the main text, and are not repeated here for brevity.

*Response of the squirmer to the external velocity field.* We use Faxén's laws[33,35,39] to evaluate the rheotactic (translational) velocity and the angular velocity of the force-free and torque-free spherical squirmer in response to $\mathbf{u}_t$ (Eq. (2)). Following Faxén's first law, the translational velocity of the force-free, spherical squirmer can be written as

$$\mathbf{v}_r = \left( \mathbf{u}_t + \frac{1}{6}\nabla^2 \mathbf{u}_t \right)_{\mathbf{r}_0}$$
$$\equiv \hat{\mathbf{e}} - \left( \mathbf{u}_f + \frac{1}{6}\nabla^2 \mathbf{u}_f \right)_{\mathbf{r}_0} + \left( \mathbf{u}_{HI} + \frac{1}{6}\nabla^2 \mathbf{u}_{HI} \right)_{\mathbf{r}_0} \qquad (9)$$

where the terms within parenthesis are evaluated at the location of the squirmer centroid $\mathbf{r}_0$. It must be remembered here that theoretically the response of the squirmer to $\mathbf{u}_s$ (Eq. (1)) cannot be evaluated using Faxén's law, as, by definition, $\mathbf{u}_s$ diverges at $\mathbf{r}_0$. But this is circumvented by noting that it is the intrinsic swimming velocity of the squirmer $\mathbf{v}_0 = v_0\hat{\mathbf{e}}$ which gives rise to $\mathbf{u}_s$ in the first place. Conversely, it can be said that $\mathbf{v}_0$ is the squirmer's response to maintain $\mathbf{u}_s$. Since Eq. (9)

is written in a non-dimensional form (non-dimensionalised by $v_0$), the first term is simply $\hat{\mathbf{e}}$ instead of $\mathbf{v}_0$. Furthermore, note that the spatial variables involved in the $\nabla^2$ operator are also non-dimensionalised by $R_d$. Using the expressions for $\mathbf{u}_f$ and $\mathbf{u}_{HI}$ (Eq. (2)) in Eq. (9), the non-dimensional $x$ and $y$ components of the translational velocity of the squirmer can be written as in Eqs. (3) and (4) respectively. Following Faxén's second law, the angular velocity of the torque-free, spherical squirmer can be written as

$$\bar{\Omega} = \frac{1}{2}\nabla \times \left( \mathbf{u}_{HI} - \mathbf{u}_f \right) \qquad (10)$$

Using Eq. (10) in $\dot{\hat{\mathbf{e}}} = \bar{\Omega} \times \hat{\mathbf{e}}$, the rate of change of the intrinsic swimming orientation of the squirmer can be written as in Eq. (5).

*Solution of the equations of motion.* The sin and cos terms within the $\sum_n$, in Eq. (5) and Eq. (3) respectively, are approximated by considering the first three terms of the corresponding Taylor series expansions. Subsequently, the infinite sums are numerically approximated using Wolfram Mathematica. The coupled ordinary differential equations, Eqs. (4) and (5), are solved numerically for $\bar{y}_0(\bar{t})$ and $\Psi(\bar{t})$ using the built-in ODE solver for initial value problems (ode45) in MATLAB. Thereafter, Eq. (3) is numerically integrated using the trapezoidal method.

## Data availability

The videomicroscopy data supporting the findings of this study have been deposited as compressed video on figshare (https://doi.org/10.4121/19657431). Raw data are available from the corresponding authors upon reasonable request.

## Code availability

MATLAB snd Mathematica scripts to solve the equations of motion based on hydrodynamic theory have been deposited on figshare (https://doi.org/10.4121/19657431). Particle tracking and PIV analyses can be recreated by any commonly used tool like PIVlab or ImageJ.

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

## Acknowledgements
C.C.M., B.V.H. and R.D. acknowledge funding from the DFG SPP 1726 "Micro-swimmers", grant number MA 6330/1-2, and C.C.M. and C.J. from the BMBF/MPG MaxSynBio initiative. R.D. also acknowledges support from IIT Hyderabad.

## Author contributions
R.D., C.M.B., B.V.H., C.J. performed experiments; R.D. and C.C.M. designed the study; R.D. performed analytical and numerical modelling; R.D., C.M.B., C.C.M., B.V.H. analyzed data and wrote the paper.

## Funding

## Competing interests
The authors report no competing interests.
