## [Peer Review File · Nature Communications]

REVIEWER COMMENTS

Reviewer #1 (Remarks to the Author):

The authors investigated the swimming behavior of active droplet microswimmers in nonuniform and highly constrained shear flow experimentally realized by a pressure-driven microchannel Poiseuille flow. They observed the dynamical behavior of the droplets depending on the flow strength, quantified by the channel flow speed averaged over the y direction. A transition from periodic upstream motion to periodic downstream motion has been identified. This is in contrast to previously reported rheotactic behavior of spherical artificial microswimmers observed in experiments close to single surfaces. The periodic droplet motion is further quantified by the dynamic droplet orientation which again modulates periodically in time. In order to disentangle the hydrodynamic and chemical origins of their findings, they measured local chemical (micelle) distribution around the droplet. The authors conclude that a pure hydrodynamic model is able to explain their results very well. The hydrodynamic model includes solid-body advection and rotation in flow, and hydrodynamic swimmer-surface interactions based on far-field approximations.

First of all, the experimental findings show nice and clear periodic trajectories which - depending on the flow rate - correspond to up- and down-stream motion. These results seem to be robust, indicated by the supplementary movie S3, although it would be good if the authors could be more quantitative: (i) how many trajectories did you observe for the different considered flow rate, and (ii) how do their dynamics depend on the initial position in the y direction in the channel. For example, can the authors show several experimentally produced phase space curves from different trajectories for a constant shear rate in a single plot (as shown in Fig. 5b for a single trajectory)? Similar phase space plots have been shown in previous works for biological microswimmers in Poiseuille flow which show similar periodic dynamics in Poiseuille flow, see e.g. ref [9].

I am, however, not convinced regarding the novelty of the observed dynamics. First, similar up- and down-stream oscillatory dynamics in Poiseuille flow, including also loops in the trajectories, has been observed by several experimental studies with biological microswimmers, see e.g. Refs [6,9,46]. This behavior is not limited to elongated swimming organisms but a general feature which stems from the coupling of self-propulsion and response to quadratic flow calculated by Faxen relations, as predicted and explained for spherical microswimmers already in Ref [25]. Indeed, most of the observed dynamics in the underlying manuscript can already be explained using the model in [25] where all types of periodic up- and downstream trajectories as shown in the underlying manuscript have already been predicted. Hence I do not agree with the statements "we will demonstrate that the active droplet swims upstream in a novel, steady oscillatory trajectory despite its spherically symmetric shape" (page 2) and "This behavior constitutes a new paradigm for rheotaxis of artificial microswimmers" (page 2) and "the droplet ... swims upstream in a novel, channel-wide oscillatory trajectory in the X-Y plane" (page 2).

The hydrodynamic model presented in the underlying manuscript is indeed similar to the one presented in [25] but it includes additional terms considering hydrodynamic interactions with the four bounding channel surfaces. However, it remains unclear how much different the model results are when the interaction with the channel walls are neglected, despite the fact that the authors claim "we demonstrate that the oscillatory rheotaxis of the droplet is primarily governed by the interaction of the finite-sized microswimmer with all four microchannel walls, and the shear flow characteristics." For me it seems that the hydrodynamic interactions with the surfaces only play a minor role, and the main dynamics can be described without them, despite the fact that the walls are so close.

In the underlying manuscript the model indicates a stable limit cycle for the pusher active droplets under consideration. The fact that hydrodynamic wall interactions of pusher microswimmers result in stable limit cycles under confined Poiseuille flow has already been described in [25], and experimentally confirmed with artificial non-spherical microswimmers [47]. The observed trend in the trajectory wavelengths with varying flow strength is again in accordance with the model presented in [25], which can be used to explain the wavelengths.

In the underlying manuscript a transition between upstream and downstream motion is observed around $u_{fa} > 0.53$, where u_{fa} is the y-averaged flow speed. The authors claim that this is "contrary to earlier theoretical predictions of downstream drift of microswimmers for $u_{fa} > 1$ using reduced order models [25, 26]." However, it should be noted that in [25] not the average flow speed has been used, but the maximum flow speed in the channel center, and that the specific up/downstream - and hence also the trapping condition - depends on the initial condition. Therefore it is not so clear how 'contrary' this is. Furthermore, it is not clear if the flow profile presented in Fig. 1b in the underlying manuscript has been obtained by averaging over the z component as well, or obtained at, for example $z=0$ (midplane).

The authors claim "The consistent inclusion of the full hydrodynamic interaction of the finite-sized microswimmer with all the confining walls, while estimating v_r and ψ , results in the recreation of the downstream drift already for $u_{af} < 1$ " (page 8). However, the authors do not show this, i.e. what happens if the hydrodynamic interactions are turned off. I suspect that the up/downstream condition does not depend much on the hydrodynamic swimmer-surface interactions for their values of α , β and γ .

Altogether, many findings in the underlying manuscript have already been identified in Ref [25], such as the fact that the swimmer direction points upstream even as it moves downstream, or the stable limit cycle behavior in ψ - γ phase space.

The authors claim (page 2) "We believe that our understanding of this new oscillatory rheotactic behavior will aid in conceptualizing and designing both in vitro and in vivo applications, like targeted cargo delivery, which involve navigation of microswimmers in confinements" and on page 8: "We strongly believe that the understanding of this new rheotactic behavior of artificial microswimmers, including the possibility of hydrodynamically trapping these, will open up wide possibilities in conceptualizing and designing practical micro-robotic applications" However, they do not support these claims by specific arguments.

What I find intriguing is the fact that the orientation dynamics can be well-described with the Faxen relations, usually applied for rigid particles. Since droplets are not rigid and the interplay of flow and surface-covered surfactants is expected to be nontrivial, it is surprising that the rigid-body-rotation works so well. Do the authors have any idea why this is the case?

One more comment: can the authors comment in more detail how they obtain the intrinsic swimming speed v_0 for different droplets? Or is it (approximately) the same for all the considered trajectories?

All in all the claim of the authors to advance the understanding of rheotaxis of spherical active particles in confined flows by 'novel' findings is not supported by their results. The significance for the scientific field remains elusive.

[Reference numbers coincide with numbers in the manuscript, except new [46]]

[6] Uppaluri et al, Biophys J 2012

[9] Junot et al, EPL 2019

[25] Zottl and Stark, PRL 2012

[46] Rusconi et al, Nature Physics 2014

Reviewer #2 (Remarks to the Author):

I report here on the manuscript entitled 'oscillatory rheotaxis of artificial swimmers in microchannels' by Dey et al.

The manuscript describes the observation and analysis of the swimming pattern of an artificial swimmer in a shear flow. The swimmer is made of an oil droplet in a surfactant solution, a system previously reported by the same group. The self-propulsion is driven by the diffusion of the oil of the swimmer towards surfactant micelles leading to a diffusophoretic motion. Among the striking properties of the propulsion in microchannels is the emergence of an oscillatory behaviour when the droplet moves on average against the imposed flow. The authors provide a detailed experimental analysis of the motion, including a very thorough description of the swimm parameters. The level of analysis of the experimental data (oscillatory behaviour, orientation of the tail of the expelled oil, angle measurements...) leads to a convincing manuscript.

I have several concerns though:

The observation of oscillatory patterns has been made already, by the authors in a different geometry and with a slightly different formulation of the mixtures used (PhysRevLett.117.048003). I find extremely puzzling that the authors do not refer to this paper nor discuss the relationship between both observations.

The generality of the observed behavior is questionable: although the authors claim that there is a link with biological rheotaxis behavior, one might ask is the observed behavior is intrinsic to this particular system. In other words, the authors do not provide values for a control parameter of the behavior that would be transferable to other systems.

I do have a problem with the 'quasi 2-d' term. The aspect ratio of the channel used is $h/2w = 1/2$ which I would not consider close to 2d. Flows around droplets in Hele-Shaw cells very quickly display 3d features. Here the situation is even closer to a 3d cell: the droplet radius of curvature is of order the channel height. I do not see the relevance of the 2d approximation. I also do not understand the parabola fit of Fig. 1b and the particle tracking performed. There must be a variation of the particle velocity along z . The flow in a rectangular channel can be analytically expressed at least with the first terms of a Taylor series. Wouldn't it be more appropriate ? This is important as these results are used later (2nd column - bottom of page 5).

I find it hard to understand how v_r is defined, especially based on how the correction is made with respect to v_0 . Here, the authors fail short on explaining this correction (even page 11). Maybe the reader would need a little help here.

Reviewer #3 (Remarks to the Author):

Dear Editor,

Thank you very much for forwarding me this manuscript. I find the results technically sound and original. A revised paper can be suitable for publication in Nature Communications.

I have major technical concerns with the work.

1. The definition of the intrinsic orientation angle Ψ of the active droplet. For a Janus particle, the orientation angle is well-defined. However, experimental extraction of this quantity for a spherical droplet is rather challenging. The authors use a technique based on image post-processing and the PIV analysis. This approach is indirect and may potentially lead to bias and systematic errors. A more direct technique is an extraction of the chemical angle Ψ_c . It appears that Ψ and Ψ_c have a similar trend, but differ by a constant. I suggest discussing the uncertainty in the angle Ψ or possibly relate the droplet orientation to Ψ_c .

2. I find the theoretical analysis quite puzzling and somewhat obscure. The authors use up to 12 image terms in the expression for the rheotactic velocity, Eq. (3)-(5). The resulting expressions are complex and provide little insight into the dynamics of the system. Since no quantitative comparison with the experiment is provided, what point is employing such an elaborate theory? Furthermore, it is not clear what is the most significant contribution in Eqs. (3)-(5) leading to the limit cycle behavior. I suggest simplifying the analysis and identifying the minimal set of terms in Eq. (2) responsible for the desired behavior. Alternatively, the author may present a more detailed comparison between theoretical and experimental results.

3. Minor remark: in the abstract, spiral-> helical

First of all, we thank the editors and all reviewers for their appreciation and detailed and constructive comments, and for highlighting points for improvement.

They have raised several concerns which we have addressed by extending and clarifying our statements in the manuscript, and, where required, by providing supplementary data and experimental details. Any changes that exceed minor rewording in the manuscript and supporting information have been highlighted in colour.

A separate point-by-point response to each reviewer follows below.

We note that, since the bibliography has changed, we follow the reference numbering of the revised manuscript where not stated otherwise.

Reviewer #1

The authors investigated the swimming behavior of active droplet microswimmers in nonuniform and highly constrained shear flow experimentally realized by a pressure-driven microchannel Poiseuille flow. They observed the dynamical behavior of the droplets depending on the flow strength, quantified by the channel flow speed averaged over the y direction. A transition from periodic upstream motion to periodic downstream motion has been identified. This is in contrast to previously reported rheotactic behavior of spherical artificial microswimmers observed in experiments close to single surfaces. The periodic droplet motion is further quantified by the dynamic droplet orientation which again modulates periodically in time. In order to disentangle the hydrodynamic and chemical origins of their findings, they measured local chemical (micelle) distribution around the droplet. The authors conclude that a pure hydrodynamic model is able to explain their results very well. The hydrodynamic model includes solid-body advection and rotation in flow, and hydrodynamic swimmer-surface interactions based on far-field approximations.

1. First of all, the experimental findings show nice and clear periodic trajectories which - depending on the flow rate - correspond to up- and down-stream motion. These results seem to be robust, indicated by the supplementary movie S3, although it would be good if the authors could be more quantitative: (i) how many trajectories did you observe for the different considered flow rate, and (ii) how do their dynamics depend on the initial position in the y direction in the channel. For example, can the authors show several experimentally produced phase space curves from different trajectories for a constant shear rate in a single plot (as shown in Fig. 5b for a single trajectory)? Similar phase space plots have been shown in previous works for biological microswimmers in Poiseuille flow which show similar periodic dynamics in Poiseuille flow, see e.g. ref [9].

Reply: With respect to the suggestion of showing several phase plots, we feel that additional data would crowd the respective figure in the main text.

Therefore, to address the reviewer's question, we have added *Supplemental Figure S5* with additional experimental data sets demonstrating similar stable limit cycles for a definite value of the imposed flow rate ($\bar{u}_f^a \sim 0.4$).

Regarding the reviewer's question about initial conditions: Given the strong wall attraction of the droplet microswimmer in a quiescent medium (see Supplemental Video S1), we cannot experimentally control the initial y position. However, the theoretical model shows that for a given \bar{u}_f^a , the steady state rheotactic trajectory, and hence the stable limit cycle, is independent of the initial position along the y -direction in the microchannel. This is shown in Figure 5b(i), in which rheotactic trajectories starting from two different y locations converge on the same stable limit cycle for a given \bar{u}_f^a .

2. I am, however, not convinced regarding the novelty of the observed dynamics. First, similar up- and down-stream oscillatory dynamics in Poiseuille flow, including also loops in the trajectories, has been observed by several experimental studies with biological microswimmers, see e.g. Refs [6,9,46]. This behavior is not limited to elongated swimming organisms but a general feature which

stems from the coupling of self-propulsion and response to quadratic flow calculated by Faxen relations, as predicted and explained for spherical microswimmers already in Ref [25]. Indeed, most of the observed dynamics in the underlying manuscript can already be explained using the model in [25] where all types of periodic up- and downstream trajectories as shown in the underlying manuscript have already been predicted. Hence I do not agree with the statements "we will demonstrate that the active droplet swims upstream in a novel, steady oscillatory trajectory despite its spherically symmetric shape" (page 2) and "This behavior constitutes a new paradigm for rheotaxis of artificial microswimmers" (page 2) and "the droplet ... swims upstream in a novel, channel-wide oscillatory trajectory in the X-Y plane" (page 2).

Reply: We would like to stress the distinction between "general oscillatory dynamics" and "physically reasonable oscillatory dynamics". While the former is indeed covered by reduced models like Zöttl 2012 ([25], now [37]), their choice of a minimal model is conceptually elegant, but includes in consequence unphysical effects (see details in reply to comment no. 3) which the authors freely acknowledge ("crashes into the wall"). We believe that an extended model replicating real-life features like finite-size and boundary effects provides considerable advancement.

We have reworded the introduction somewhat to make our claims of novelty more precise. As the reviewer notes, general oscillatory upstream swimming has been predicted analytically. It has also been observed in biological systems — however, here the causality is obfuscated by inherent asymmetry, bio-complexity and the general noise and reproducibility challenges in biophysics, as we have discussed in the manuscript's literature review. That is why we use an experimental artificial swimmer designed to have maximal spherical symmetry, and we are not aware of any such previous experiments that show upstream oscillation with a regularity that allows for a detailed quantitative comparison to theoretical modelling.

We thank the reviewer for bringing Rusconi 2014 ("[46]", now new ref. [8] in the main text) to our attention: we believe their findings actually highlight the need for quantitative experiments combined with theoretical modelling in artificial swimmer paradigms of high symmetry. Especially in flagellated biological swimmers, it is a highly non-trivial task to separate oscillatory instabilities based on the swimmer geometry (cf. Lauga and Berg's seminal work on *E. coli* swimming in circles at interfaces, Biophys J. 90(2), 400 (2006)) from those caused by external influences like imposed flow and boundaries. In fact, Rusconi 2014 do not analyse the individual dynamics, and their trapping effect is a very different phenomenon, as it refers to accumulation in high shear regions near the channel walls, mediated by the swimmer geometry. They even explicitly point out that this rheotactic accumulation is supposed to vanish for spherically symmetric objects. Similarly, Junot 2019 ([9], now [10]) features shape-dependent Jeffery orbits and no wall effects.

3. The hydrodynamic model presented in the underlying manuscript is indeed similar to the one presented in [25] but it includes additional terms considering hydrodynamic interactions with the four bounding channel surfaces. However, it remains unclear how much different the model results are when the interaction with the channel walls are neglected, despite the fact that the authors claim "we demonstrate that the oscillatory rheotaxis of the droplet is primarily governed by the interaction of the finite-sized microswimmer with all four microchannel walls, and the shear flow characteristics." For me it seems that the hydrodynamic interactions with the surfaces only play a minor role, and the main dynamics can be described without them, despite the fact that the walls are so close.

Reply: We respectfully disagree with the reviewer. To prove the importance of those hydrodynamic interactions, we now additionally include hypothetical rheotactic trajectories missing these features in *Fig. 5(a)*, and the corresponding stable limit cycles in *Fig. 5b(ii)*, in the revised manuscript.

First, a grey line which represents the trajectory for an active particle in a pressure-driven flow without any hydrodynamic interaction with the walls of the microchannel (no contribution from the singularities and their image systems in Eqs. 3-5).

Second, a green line which represents the rheotactic trajectory for a point-sized, pusher-type microswimmer in a pressure-driven flow (i.e. only the contribution from the force-dipole singularity and its image system has been considered in Eqs. 3-5).

Note that these are similar to the two cases discussed in Zöttl 2012. Of course these simplified/reduced

order models generate an oscillatory rheotactic trajectory, and hence stable limit cycles, as can be expected from such non-linear dynamical models. However, these models differ from the experimentally observed oscillatory rheotactic dynamics in a number of essential features, which are satisfactorily addressed only by the extended theoretical model described in our manuscript.

An omission of the hydrodynamic interactions, especially the effect of the finite size of the microswimmer, will lead to aphysical predictions of the rheotactic behaviour of artificial microswimmers which can be detrimental for designing micro-robotic applications:

First, the reduced order models do not capture the shape of the experimentally observed rheotactic trajectory for a definite imposed flow rate (compare Figs. 3a and 5a). Second, these simplified models significantly underestimate the intrinsic orientation (Ψ) of the microswimmer over the majority of its rheotactic trajectory (see Figs. 5b(i) and (ii)). Third, they do not correctly capture the variation in the rheotactic velocity of the microswimmer over its oscillatory trajectory, which is well addressed by our model (compare Figs. 1g and 5c).

From such comparison between experimental and theoretical results, it is clear that the rheotactic dynamics of a finite-sized microswimmer in a narrow microchannel is strongly influenced by its hydrodynamic interaction with the confinement walls and its interaction with the imposed shear flow.

4. In the underlying manuscript the model indicates a stable limit cycle for the pusher active droplets under consideration. The fact that hydrodynamic wall interactions of pusher microswimmers result in stable limit cycles under confined Poiseuille flow has already been described in [25], and experimentally confirmed with artificial non-spherical microswimmers [47]. The observed trend in the trajectory wavelengths with varying flow strength is again in accordance with the model presented in [25], which can be used to explain the wavelengths.

Reply: Indeed, a stable limit cycle is a feature of any robust oscillatory behaviour. Our reason for including and discussing limit cycles is that our calculated and measured limit cycles illustrate the rheotactic dynamics of a microswimmer in a narrow microchannel that is not captured by less detailed hydrodynamic models, as discussed above. A physically consistent understanding of rheotaxis in confinements is of primary importance to real-life situations and applications.

Regrettably, the reviewer’s reference ”[47]” appears to have gotten lost somewhere — we googled but couldn’t find anything beyond the studies already covered in our bibliography.

5. In the underlying manuscript a transition between upstream and downstream motion is observed around $u_f^a > 0.53$, where u_f^a is the y-averaged flow speed. The authors claim that this is ”contrary to earlier theoretical predictions of downstream drift of microswimmers for $u_f^a > 1$ using reduced order models [25, 26].” However, it should be noted that in [25] not the average flow speed has been used, but the maximum flow speed in the channel centre, and that the specific up/downstream - and hence also the trapping condition - depends on the initial condition. Therefore it is not so clear how ’contrary’ this is. Furthermore, it is not clear if the flow profile presented in Fig. 1b in the underlying manuscript has been obtained by averaging over the z component as well, or obtained at, for example $z = 0$ (midplane).

Reply: We agree that Zöttl 2012, in fact, only states an implicit up/downstream condition, as well as an explicit, but considerably higher threshold for tumbling and swinging motion — even considering the centerline velocity it is much greater than the intrinsic swimming velocity of the microswimmer. We have now removed the respective statement from the manuscript.

Concerning the possible z averaging of our measured flow profile: We used high magnification microscopy, trained on the midplane with a shallow focal depth of $\approx 1.5\mu\text{m}$ for PIV. Thus, the profile is obtained at $z = 0$ (the plane of motion of the active droplet) with reasonable accuracy. We have added the respective details to the Methods section (*p. 9; line 604*). Note that we use this velocity field to evaluate the orientation (Ψ) of the active droplet (*see the Methods section p. 10; line 650*). However, for the theoretical model we use the exact velocity profile for a pressure-driven flow through a rectangular cross-section microchannel.

Figure 1: Theoretical predictions for rheotaxis of a weak pusher (column (a): $\alpha = 0.16$, $\beta = 0.26$, $\gamma = -0.08$; identical to that used in the manuscript that corresponds to the experimental droplet swimmer) and a relatively strong pusher (column (b): $\alpha = 0.75$, $\beta = 0.26$, $\gamma = -0.08$) through a narrow microchannel for identical values of $\bar{u}_f^a (= 0.42)$. Rows (i) and (ii) show the variations of the intrinsic swimming orientation Ψ (non-dimensionalised by π) and the non-dimensional rheotactic velocity \bar{v}_r , respectively over the corresponding trajectories. A strong pusher tends to slide more along the walls at a relatively higher rheotactic velocity, and exhibits smaller changes in its intrinsic orientation. Accordingly, the wavelength of the oscillatory upstream trajectory for strong pushers is significantly larger than that for finite-sized weak pushers.

6. The authors claim "The consistent inclusion of the full hydrodynamic interaction of the finite-sized microswimmer with all the confining walls, while estimating v_r and ψ , results in the recreation of the downstream drift already for $u_f^a < 1$ " (page 8). However, the authors do not show this, i.e. what happens if the hydrodynamic interactions are turned off. I suspect that the up/downstream condition does not depend much on the hydrodynamic swimmer-surface interactions for their values of α , β and γ .

Reply: We thank the reviewer for suggesting a comparison to a model without hydrodynamic interactions; we have now illustrated the importance of hydrodynamic interactions in dictating the oscillatory upstream rheotaxis by comparing to reduced order models without, or including not fully realised hydrodynamic interactions (see our reply to comment no. 3). Additionally, we have now also checked, using our theoretical framework, that on switching off the hydrodynamic interactions, the threshold value of \bar{u}_f^a for the upstream to downstream switching increases by a significant 49%. This implies that if hydrodynamic interactions are neglected, one might assume an upstream rheotaxis for values of $\bar{u}_f^a \leq 1$ ($\sim 0.82 - 1.22$), where, as we are able to confirm in experiments, the microswimmer is going to drift downstream. Therefore, switching off the hydrodynamic interactions does affect the upstream/downstream condition for our values of α , β , γ .

Furthermore, the rheotactic dynamics indeed depend on the hydrodynamic interactions, and hence, on the parameters α , β , γ . To prove our point we have added numerically calculated trajectories for changed parameter values (corresponding to pullers, i.e. to negative values of α) to the *Supplemental Figure S4*.

We add one more example for changed α to this response, the numerically calculated trajectories for a strong pusher, compared to the weak pusher used in the manuscript that corresponds to the experimental droplet swimmer (see Figure 1).

7. Altogether, many findings in the underlying manuscript have already been identified in Ref [25],

such as the fact that the swimmer direction points upstream even as it moves downstream, or the stable limit cycle behaviour in ψ - y phase space.

Reply: Here we refer to our discussions above on how and why reduced order models do not sufficiently capture realistic rheotactic behaviour – we would like to note, additionally, that Zöttl 2012 is a pure theory paper and that we provide quantitative experimental validation of these features as well.

8. The authors claim (page 2) "We believe that our understanding of this new oscillatory rheotactic behaviour will aid in conceptualising and designing both in vitro and in vivo applications, like targeted cargo delivery, which involve navigation of microswimmers in confinements" and on page 8: "We strongly believe that the understanding of this new rheotactic behaviour of artificial microswimmers, including the possibility of hydrodynamically trapping these, will open up wide possibilities in conceptualising and designing practical micro-robotic applications" However, they do not support these claims by specific arguments.

Reply: We are happy to provide a more detailed discussion — we feel that the best place for it would be the end of the conclusion and have added the respective statements. Briefly, we argue that practical long-time trapping of self-motile agents is actually quite hard to do, but here feasible via a robust feedback loop, and that having a realistic picture of the orientation and dynamics at interfaces is invaluable in designing applications — after all, that is where the majority of interactions with the environment is expected to happen.

9. What I find intriguing is the fact that the orientation dynamics can be well-described with the Faxén relations, usually applied for rigid particles. Since droplets are not rigid and the interplay of flow and surface-covered surfactants is expected to be nontrivial, it is surprising that the rigid-body-rotation works so well. Do the authors have any idea why this is the case?

Reply: Here, we can think of two arguments: For one, the system has a low capillary number, such that the droplets can be treated as non-deformable with high accuracy. Second, the squirmer description is agnostic regarding the mechanism of actuation, which is why it is so popular as a general model for all kinds of swimmers. A droplet driven by an interfacial tension discontinuity and a rigid phoretic particle driven by a phoretic slip velocity produce both similar hydrodynamics in the external phase, which is the level of granularity at which we apply the Faxén relations.

10. One more comment: can the authors comment in more detail how they obtain the intrinsic swimming speed v_0 for different droplets? Or is it (approximately) the same for all the considered trajectories?

Reply: The intrinsic swimming speed used in the manuscript was determined from separate experiments on droplets in a quiescent medium in z confinement, with similar droplet size and fuel concentration. We have added experimental details to the methods (*p. 10; line 614*) and an error estimate to the main manuscript (*p. 2; line 151*).

11. All in all the claim of the authors to advance the understanding of rheotaxis of spherical active particles in confined flows by 'novel' findings is not supported by their results. The significance for the scientific field remains elusive.

We have been more precise about what is specifically novel in our responses to all reviewers, and have stated our claims more unequivocally. To summarise, we are not aware of previous experimental proof of oscillatory motion based on purely spherically symmetric *artificial* microswimmers in pressure-driven flows through a microchannel. Since torques causing general (no flow) oscillatory motion or helical/circular swimming are also a consequence of built-in broken symmetries, which are common in biological and many artificial swimmers (cf. e.g. PRL 126, 088003 (2021); Integr Comp Biol, 36(6), 608 (1996); PRL 110, 198302 (2013); PRL 117, 048003 (2016)), we believe it is important to provide unequivocal experimental proof with reproducible, high-resolution data for imposed flow and hydrodynamic interaction driven oscillatory rheotaxis. We do not believe that numerical works like Zöttl 2012 would lessen that novelty.

We further believe that the explicit inclusion of the hydrodynamic interaction with walls is a significant

improvement. The MPCD simulations in Zöttl 2012 point in that direction (compare eqn 6 and 8), and we are able to prove by providing experimental and analytical data that this HD interaction significantly changes the behaviour at walls, the trajectory dynamics and moreover can trace these back to the respective HD singularities. What active agents do at interfaces is of high practical interest to biophysicists and smart materials engineers, so modelling and analysing effects that notably affect residence dynamics should bring the field forward. We have expanded our discussion to that effect.

References cited by the reviewer, with original submission numbering: [6] Uppaluri et al, Biophys J 2012; [9] Junot et al, EPL 2019; [25] Zöttl and Stark, PRL 2012; [46] Rusconi et al, Nature Physics 2014

Reviewer #2

I report here on the manuscript entitled 'oscillatory rheotaxis of artificial swimmers in microchannels' by Dey et al.

The manuscript describes the observation and analysis of the swimming pattern of an artificial swimmer in a shear flow. The swimmer is made of an oil droplet in a surfactant solution, a system previously reported by the same group. The self-propulsion is driven by the diffusion of the oil of the swimmer towards surfactant micelles leading to a diffusiophoretic motion. Among the striking properties of the propulsion in microchannels is the emergence of an oscillatory behaviour when the droplet moves on average against the imposed flow. The authors provide a detailed experimental analysis of the motion, including a very thorough description of the swim parameters. The level of analysis of the experimental data (oscillatory behaviour, orientation of the tail of the expelled oil, angle measurements...) leads to a convincing manuscript.

I have several concerns though:

1. The observation of oscillatory patterns has been made already, by the authors in a different geometry and with a slightly different formulation of the mixtures used (PhysRevLett.117.048003). I find extremely puzzling that the authors do not refer to this paper nor discuss the relationship between both observations.

Reply: The active droplets used in our previous work (PRL 117, 048003) are chemically different from the present droplets in that the previous ones were nematic isomers of the same oil molecule. For nematic active droplets, the oscillation emerges from the interaction between the spontaneously broken symmetry around the droplet and the elastic properties of the liquid crystal. Such oscillations do not require the effects of confinement and/or external flow. However, in the present study, the oscillations of the isotropic active droplets (no liquid crystal effects are present here) emerge from the continuous reorientation of the droplet due to the angular velocity induced by the pressure-driven flow and the hydrodynamic interaction with the microchannel walls. Since the physical mechanism causing the oscillation of the isotropic active droplets is fundamentally different from those previously observed for liquid crystal droplets, we opted for omitting the references, but we have included and addressed them now to highlight the difference, as suggested by the reviewer (*see pp. 2-3, lines 184-189 of the revised manuscript*).

2. The generality of the observed behavior is questionable: although the authors claim that there is a link with biological rheotaxis behavior, one might ask is the observed behavior is intrinsic to this particular system. In other words, the authors do not provide values for a control parameter of the behavior that would be transferable to other systems.

Reply: The fact that a purely long-range hydrodynamic model (sans any chemical coupling), based on the Stokes singularities, classical method of images, and Faxén's laws, captures the essential features of the observed rheotactic dynamics indicates the generality of the behaviour: i.e., any finite-sized, spherical, pusher-type microswimmer (like the majority of the artificial microswimmers) will undergo similar rheo-

tactic behaviour in a narrow microchannel. The rheotactic behaviour of other types of microswimmers, e.g. pullers, in a microchannel can be also qualitatively predicted using the model considering α and β as the tunable parameters. This is applicable to the possible rheotactic behaviour of biological microswimmers in confinements. We now show such a behaviour in the Supplementary material (*see Supplemental Fig. S4*), with a discussion of recent literature. Of course, in reality, there will be differences between the predicted behaviour and the actual rheotaxis of biological microswimmers stemming from factors like biocomplexity in shape or propulsion mechanism.

However, exactly these types of hydrodynamic models are widely used (see e.g. ref [6]; PRL 101, 038102, (2008); Phys. Fluids 26, 071902, (2014); JFM 806, p. 35-70, (2016); PRL 123, 248102, (2019)) for the generic description of biological swimmers, assuming that biocomplexity does not affect the validity of the model. Thus, our work provides the first comparative study between experiments based on a simple model system and the long-range hydrodynamic theory to evaluate the efficacy of these purely hydrodynamic approaches in addressing the rheotaxis of microswimmers in confinements. Insights from this study are definitely applicable to other artificial micro-robotic active agents besides active droplets, and even biophysical systems, albeit in a qualitative manner.

3. I do have a problem with the 'quasi 2-d' term. The aspect ratio of the channel used is $h/2w = 1/2$ which I would not consider close to 2d. Flows around droplets in Hele-Shaw cells very quickly display 3d features. Here the situation is even closer to a 3d cell: the droplet radius of curvature is of order the channel height. I do not see the relevance of the 2d approximation. I also do not understand the parabola fit of Fig. 1b and the particle tracking performed. There must be a variation of the particle velocity along z . The flow in a rectangular channel can be analytically expressed at least with the first terms of a Taylor series. Wouldn't it be more appropriate? This is important as these results are used later (2nd column - bottom of page 5).

Reply: We thank the reviewer for this suggestion. In the revised manuscript, we do not use the quasi-2D approximation anymore, but have now used the exact velocity profile for a pressure-driven flow through a microchannel with a rectangular cross-section (*Eq. 2*). We have re-derived the equations of motion using Faxén's laws accordingly (*Eqs. 3-5*), and have re-plotted the theoretical results in *Fig. 5* by solving these equations. The resulting changes do not alter our previous conclusions; in fact, the consideration of the exact velocity profile provides a better comparison between the shapes and velocities of the experimental and theoretical rheotactic trajectories.

We note that the active droplet centroid is constrained to move in the $X - Y$ plane ($z_0 = 0$ throughout). We have used high magnification microscopy to measure the flow field in the immediate vicinity of the droplet centroid, i.e. adjacent to the droplet at the mid-plane of the channel ($z = 0$ plane). Since we perform this PIV analysis using a $20\times$ lens with numerical aperture $NA = 0.7$, and given that the pixel size of the sensor is $4.3\mu\text{m}$, we conclude that the depth of focus of our measurement is $\approx 1.5\mu\text{m}$. Hence, we are quite certain that we indeed do the PIV measurement in the plane of motion of the active droplet centroid. We use this velocity field to evaluate the orientation (Ψ) of the active droplet (*see the Methods section p. 10, line 650*). The velocity profile in Fig. 1b shows the variation of the x-component of the imposed pressured-driven flow velocity over y in the $z = 0$ plane. The velocity profile is obtained from a similar PIV analysis, but performed over a region further away from the droplet. Here, $u_x(y) \equiv \langle u_x(x, y, z = 0) \rangle_x$, where $u_x(x, y, z = 0)$ is the velocity field obtained from the PIV analysis. The purpose of Figure 1b is to show that the uni-directional flow in the plane of oscillation of the active droplet has a quadratic dependence in y , as is expected in the leading order from the classical analytical analysis. Furthermore, note that for the present geometry, mathematically the average flow velocity calculated from the exact analytical solution $\int_{-w}^w \int_{-h}^h u_x(y, z) dz dy / 4hw \approx \int_{-w}^w u_x(y, z = 0) dy / 2w$.

4. I find it hard to understand how v_r is defined, especially based on how the correction is made with respect to v_0 . Here, the authors fail short on explaining this correction (even page 11). Maybe the reader would need a little help here.

Reply: \vec{v}_r is the rheotactic swimming velocity of the active droplet tracked in the laboratory reference

frame. In other words, it is the resulting velocity with which the active droplet navigates the microchannel considering its own activity, the effects of the imposed pressure-driven flow, and the hydrodynamic interactions with the microchannel walls.

\bar{v}_0 is the intrinsic swimming velocity of the active droplet, where the rheotactic speed and orientation have been corrected for the translational and rotational effects of the imposed flow and the hydrodynamic interactions with the microchannel walls at the instantaneous location of the droplet. $|\bar{v}_0| \equiv v_0$ is essentially the intrinsic swimming speed of the active droplet in an unbounded quiescent medium. We have now rewritten our explanations both in the main text (p. 2) and in the methods section (p. 10), and we hope it is more clear like that.

Reviewer #3

Dear Editor,

Thank you very much for forwarding me this manuscript. I find the results technically sound and original. A revised paper can be suitable for publication in Nature Communications.

I have major technical concerns with the work.

1. The definition of the intrinsic orientation angle Ψ of the active droplet. For a Janus particle, the orientation angle is well-defined. However, experimental extraction of this quantity for a spherical droplet is rather challenging. The authors use a technique based on image post-processing and the PIV analysis. This approach is indirect and may potentially lead to bias and systematic errors. A more direct technique is an extraction of the chemical angle Ψ_c . It appears that Ψ and Ψ_c have a similar trend, but differ by a constant. I suggest discussing the uncertainty in the angle Ψ or possibly relate the droplet orientation to Ψ_c .

Reply: We indeed use the rheotactic velocity vector, i.e. the tracked velocity of the active droplet in the laboratory reference frame, and the measured velocity field in the immediate vicinity of the droplet in the plane of oscillation (using PIV) to calculate the intrinsic orientation of the droplet Ψ (*methods section p. 10, line 650*). Of course, PIV and tracking analyses inherently introduce some measurement errors. However, note that such measurements within our specific geometry do not result in biased or systematic error (an error having a non-zero mean). We have now estimated the uncertainty in the measurement of Ψ from our technique to be $\approx 11\%$, which is reasonable. We now also mention this in the caption to Fig. 3 (p. 5)

Conversely, the chemical angle Ψ_c measured from the fluorescent microscopy data, is systematically underestimated because the ambient flow advects the chemical trail. The flow field in the immediate vicinity of the droplet advects the filled micelle trail downstream. Consequently, the point of maximum fluorescence intensity near the rear stagnation point, is always slightly displaced to the left (right), relative to the upstream swimming direction in the $X - Y$ plane, when the droplet is oriented in a positive (negative) sense relative to $+\hat{x}$. So, we always obtain a relatively smaller angle for Ψ_c compared to Ψ . We have illustrated this now with an additional *figure (S3, also copied here) in the supplementary material* (note that the illustrated difference between angles is exaggerated in the figure for the sake of clarity).

Therefore, for the present system Ψ is a more accurate estimation of the intrinsic swimming direction than Ψ_c . We still include our estimate of Ψ_c in the manuscript, as it provides general insight to the behaviour of the chemical trail during the rheotaxis of artificial microswimmers. Importantly, Ψ and Ψ_c show similar trend and are also quantitatively comparable. Furthermore, we can check that the filled micelle trail is not distorted during rheotaxis to trigger secondary interactions (see ref. 25) with the swimming droplet, and thereby interfering with the rheotactic dynamics.

2. I find the theoretical analysis quite puzzling and somewhat obscure. The authors use up to 12 image terms in the expression for the rheotactic velocity, Eq. (3)-(5). The resulting expressions are complex and provide little insight into the dynamics of the system. Since no quantitative comparison

Figure 2: Definition of Ψ_c from the fluorescent microscopy images.

with the experiment is provided, what point is employing such an elaborate theory? Furthermore, it is not clear what is the most significant contribution in Eqs. (3)-(5) leading to the limit cycle behaviour. I suggest simplifying the analysis and identifying the minimal set of terms in Eq. (2) responsible for the desired behaviour. Alternatively, the author may present a more detailed comparison between theoretical and experimental results.

Reply: We represent the flow field generated by the spherical, axi-symmetric, droplet microswimmer by an equivalent squirmer model comprising of the superposition of three Stokes' singularities- a positive force dipole, a source dipole, and a source quadrupole. Note that similar combinations are also used in recent literature for swimmers near walls/obstacles in quiescent media (see refs. [28], [31]). These three singularities physically represent three important aspects of the microswimmer: The force dipole represents the effect of the net force-free, pusher-type swimming of the active droplet; the source dipole represents the effect of the finite size of the spherical droplet microswimmer; and the source quadrupole captures the hydrodynamics close to the microswimmer. Next, we evaluate the image flow field for each of these three singularities for all the four walls of the microchannel to consistently account for the hydrodynamic interaction with the confinement walls. Omission of any one of the three singularities, and its associated images, fails to capture the experimentally obtained rheotactic dynamics. In the revised manuscript, we have added two more rheotactic trajectories in Fig. 5(a), and the corresponding stable limit cycles in Fig. 5b(ii) for illustration. The first, grey line represents the trajectory for an active particle in a pressure-driven flow without any hydrodynamic interaction with the walls of the microchannel (no contribution of the singularities and their image systems in Eqs. 3-5). The second, green line represents the rheotactic trajectory for a point-sized, pusher-type microswimmer in a pressure-driven flow (i.e. only the contribution of the force-dipole singularity and its image system has been considered in Eqs. 3-5). These two trajectories show that, while a reduced order model replicates oscillations, it overwhelmingly fails to capture the essential features of the experimentally observed oscillatory rheotactic dynamics.

While similar reduced order models without the two higher order singularities (the grey and green lines in Fig. 5) were used before to theoretically predict general oscillations (see refs. [37], [38]), the exact dynamics are only satisfactorily captured by the extended theoretical model presented here (compare Figs. 3a and 5a, and also see Fig. 5b).

Then, to consistently account for the hydrodynamic interactions images at all the four walls of the microchannel should be considered. Actually, to truly satisfy the no-slip boundary condition at all the microchannel walls, an infinite number of images for each singularity should be considered (refs. [32], [33]). However, we have considered here only one image system per singularity, per wall, as this is sufficient to capture the essential features of the rheotactic dynamics. So, in a sense the theoretical model presented here is a reduced order model already, and we can't think of a consistent way to make do with less than 12 images.

We would also like to stress that we do make direct comparisons between the experimental and theoretical results. We compare the experimentally obtained variation in the intrinsic swimmer orientation (Ψ) over the oscillatory upstream rheotactic trajectory (Fig. 3a) with the predictions of the theoretical model (Fig. 5a). We also compare the experimentally and theoretically obtained stable limit cycles for a definite imposed flow rate (Fig. 5b(i)). Importantly, we explain the physical mechanisms behind the experimentally observed variations in Ψ and the rheotactic velocity (v_r) during the oscillatory upstream rheotaxis with the

help of the inferences drawn from the theoretical results (Fig. 5(b)-(e)).

3. Minor remark: in the abstract, spiral→ helical

Reply: We have accordingly reworded the beginning of the abstract, it was indeed confusing.

REVIEWER COMMENTS

Reviewer #1 (Remarks to the Author):

The authors addressed the issues raised by me and other reviewers, and provided an updated manuscript and updated Supplemental Material. However, there are still some important points which need to be considered:

The authors now made a bit more clear what the effects of the hydrodynamic surface-swimmer interactions have on the swimmer trajectories. As far as I understand it now, the force-dipole terms allow the swimmers to spend more times closer to the boundaries, where the flow velocities are weaker, which eventually influence the critical flow strength dividing upstream/downstream motion. However, this has not been made sufficiently clear in the manuscript.

I still do not agree with some of the 'novelty' aspects mentioned in the paper. For example, on page 2 line 160 the authors state: "On actuating the pressure-driven flow in the direction opposite to the droplet motion, it swims upstream in a novel, channel-wide oscillatory trajectory in the X-Y plane". It has been demonstrated theoretically [37,50,51] and in experiments with biological [6] and artificial swimmers [51], that spherical and non-spherical active swimmers oscillate oriented upstream in microchannel flow. This is a generic feature of activity together with the non-uniform shear flow, which is stabilized by a hydrodynamic interaction of a pusher swimmer with surfaces [37]. What I think is new in the underlying study, is the very nice experimental realization with the spherical active droplets. However, I disagree that the upstream oscillation phenomenon itself is novel.

I still think that the theoretical model presented in the manuscript has not been equitably put into context with previous models. The emergence of a stable upstream oscillation, and the transition between upstream and downstream oscillation has been discussed already in Ref [37] using activity, Faxen relation, and the method of images for pusher/puller microswimmers to include hydrodynamic swimmer-surface interactions, as it has been done in the underlying manuscript. It already explains the general feature of the observed dynamics in the underlying manuscript. Here the authors include more higher-order terms for these hydrodynamic swimmer-surface interactions. While these terms do not modify the overall qualitative behavior of the oscillatory dynamics, the authors showed that including them allows for a better quantitative comparison with the experiments. Therefore, I do not agree with the statement "Thus, while reduced order descriptions replicate oscillations in general, they do not capture the essential features of the experimentally observed oscillations, while the extended model does so successfully." Also, the authors do not acknowledge that these 'reduced order models' have been developed and used in previous works to explain oscillatory upstream dynamics of swimmers in microchannel flow. For example in the Discussion Section the authors state that "Using phase portraits, we illustrate that this model features a robust limit cycle" - which has been demonstrated already in [37] using the squirmer model.

In the following I have more comments regarding the authors' answer to my initially raised points 1. - 11.

Comment to 1.: Thank you for clarifying this point. The Supplementary Figure S5 shows nicely the robustness of the phase space dynamics. Still the authors did not answer my question how many trajectories they observe for the different flow rates.

Comment to 2.: As pointed out already, I agree that the experimental system in the underlying manuscript is very nice and presents valuable new results. I disagree however

with the novelty aspect regarding the oscillatory dynamics, which has been seen in other theoretical and experimental works.

Comment to 3.: As stated above, of course I agree that including the hydrodynamic interactions is important when comparing quantitatively to the experiments, and in Ref [37] hydrodynamic interactions have been shown to be important to determine the stability of the oscillatory dynamics. I am just not convinced, that the extended model adds fundamental new features. Yes, the shape of the oscillatory trajectories are different when adding more hydrodynamic interaction terms. This is related to the fact that the extra terms shift the 'critical' flow strength which separates the upstream to the downstream trajectories, and hence the shape of the trajectories, the specific values of $\psi(t)$, and the rheotactic velocity at a specific flow rate, but do not change the essential characteristics of the oscillations.

Comment to 4.: I am sorry I missed to include the reference. It is Caldag and Yesilyurt, Journal of Fluids and Structures 90, 164, 2019; where the upstream oscillation of artificial pusher and puller swimmers in cylindrical pipe flow has been investigated, and stable upstream oscillations observed, and experiments and simulations compared. See reference [51] below.

Comment to 5.: Thank you for the clarification regarding the flow profile measurement.

Comment to 6.: I think it is good that the authors showed more explicitly how the hydrodynamic interactions influence the upstream/downstream transition and the shape of the oscillatory trajectories. While this has not been done systematically, it still gives a good impression how the hydrodynamic interactions are able to influence the upstream swimming condition.

Comment to 7.: See my comments above.

Comment to 8.: Thank you for this extra discussion. As stated above, I would be cautious naming it a 'new rheotactic behavior'.

Comment to 9.: Yes I see why it makes sense to model the droplet as a squirmer in a quiescent fluid. However, one could think that external shear flow will induce flow of the surfactants at the droplet surface which influence the droplet swimming orientation, in contrast to phoretic swimmers with an intrinsic orientation.

Comment to 10.: Thank you for the clarification.

Comment to 11.: I agree that the verification of the theoretically proposed stable upstream oscillations, as reported in [37], is a valuable and important aspect of the manuscript. I would just be more cautious talking about novel effects, and about proposing a new theoretical model. For me the 'novel' aspect in the paper is the demonstration that the specific hydrodynamic interactions of the active droplet with the boundaries influence the upstream swimming condition and shape of trajectories, and not the oscillatory upstream phenomenon itself. However, this has not been stated clearly in the manuscript. Also, several described phenomena in the manuscript have been found theoretically in [37], but not referred to in the manuscript.

References [6], [37] as in updated manuscript, and Refs [50], [51] used in this report:
[6] Uppaluri et al, Biophysical Journal 2012.

[37] Zottl Stark PRL 2012.

[50] Costanzo et al 2012 J. Phys.: Condens. Matter 24 065101.

[51] Caldag and Yesilyurt, Journal of Fluids and Structures 90, 164, 2019

Reviewer #2 (Remarks to the Author):

I report here on the resubmitted manuscript by Dey et al. In my initial comments, I had three major concerns: (i) the relationship of the results to previously described systems by the same group, (ii) the generality of the results and (iii) a technical comment on the 2d approximation made.

First of all, I am happy to see that the authors addressed these comments and that - regarding my third point - the use of a 3d description of the flow profile improves the comparison between the model and the experiments. Regarding the points (i) and (ii) there is a small inconsistency in claiming generality of the oscillatory dynamics on the one hand and arguing that the formulation used in a previous paper do not relate to the observed behaviour. So I remain a little puzzled by these statements. However, I do not think it is necessary to be too picky about this. In the end, the manuscript is a great combination of experimental results on an original system, thorough modeling and interesting discussion and I would support publication.

Reviewer #3 (Remarks to the Author):

Dear Editors,

The authors addressed my first comment on the definitions of two angles. However, I don't feel that the answer to question 2 on the choice of the truncated model is satisfactory. It is hard to believe that one needs 12 terms in the expansion for the occurrence of a limit cycle. I am pretty sure that a much simple system can capture a limit cycle. Furthermore, the comparison with the experiment shown in Fig. 5b is somewhat qualitative. Thus, a much simpler minimal model would provide a better insight than the complicated one.

We thank all three reviewers for again reading our manuscript in-depth, and for their engagement in our topic. We have discussed the remaining concerns of all reviewers, and made additional adjustments to the manuscript in response, with a detailed explanation below. The changes from the second round are now marked in blue in the manuscript. We have not opted for a full point-by-point response in all places, as some comments would overlap, and request Reviewer #1 to primarily refer to our explanations following their introductory remarks. Please note that we have added three additional references, and changed the order of others. We have chosen to use the numbering of references cited in our response as it appears in the updated bibliography. For ease of identification, we have added author-year identifiers.

Reviewer # 1 (Remarks to the Author):

The authors addressed the issues raised by me and other reviewers, and provided an updated manuscript and updated Supplemental Material. However, there are still some important points which need to be considered:

The authors now made a bit more clear what the effects of the hydrodynamic surface-swimmer interactions have on the swimmer trajectories. As far as I understand it now, the force-dipole terms allow the swimmers to spend more times closer to the boundaries, where the flow velocities are weaker, which eventually influence the critical flow strength dividing upstream/downstream motion. However, this has not been made sufficiently clear in the manuscript.

I still do not agree with some of the 'novelty' aspects mentioned in the paper. For example, on page 2 line 160 the authors state: "On actuating the pressure-driven flow in the direction opposite to the droplet motion, it swims upstream in a novel, channel-wide oscillatory trajectory in the X-Y plane". It has been demonstrated theoretically [37,50,51] and in experiments with biological [6] and artificial swimmers [51], that spherical and non-spherical active swimmers oscillate oriented upstream in microchannel flow. This is a generic feature of activity together with the non-uniform shear flow, which is stabilized by a hydrodynamic interaction of a pusher swimmer with surfaces [37]. What I think is new in the underlying study, is the very nice experimental realization with the spherical active droplets. However, I disagree that the upstream oscillation phenomenon itself is novel.

I still think that the theoretical model presented in the manuscript has not been equitably put into context with previous models. The emergence of a stable upstream oscillation, and the transition between upstream and downstream oscillation has been discussed already in Ref [37] using activity, Faxen relation, and the method of images for pusher/puller microswimmers to include hydrodynamic swimmer-surface interactions, as it has been done in the underlying manuscript. It already explains the general feature of the observed dynamics in the underlying manuscript. Here the authors include more higher-order terms for these hydrodynamic swimmer-surface interactions. While these terms do not modify the overall qualitative behavior of the oscillatory dynamics, the authors showed that including them allows for a better quantitative comparison with the experiments. Therefore, I do not agree with the statement "Thus, while reduced order descriptions replicate oscillations in general, they do not capture the essential features of the experimentally observed oscillations, while the extended model does so successfully." Also, the authors do not acknowledge that these 'reduced order models' have been developed and used in previous works to explain oscillatory upstream dynamics of swimmers in microchannel flow. For example in the Discussion Section the authors state that "Using phase portraits, we illustrate that this model features a robust limit cycle" - which has been demonstrated already in [37] using the squirmer model.

Reply:

The reviewer raised our awareness that our claims of novelty might be loosely termed and could be misinterpreted. Indeed, we would not claim that upstream oscillations themselves are a novelty (see e.g. refs. Uppaluri 2012, Junot 2019 and Mathijssen 2019 [6,11,12]). We have rephrased several sentences throughout the manuscript to identify our key findings (as explained in more detail below) unambiguously.

The foundational analytical framework enabling oscillation as such is well documented in previous works, like that of Zöttl 2012 [now ref. 29], which models the problem of an idealized, pusher-type point particle. In contrast, we aimed for the direct comparison of an *experimentally quantifiable* microswimmer to an *analytical hydrodynamic model* accounting for finite size swimmers in strong confinement.

Discussing the reviewer's comments, we found that a short summary comparing the two systems and motivating our approach was indeed lacking in the introduction, and it would clarify our intentions to contextualise our model early on, which we have done now (p. 2, line 101).

We regarded the change in near-wall dynamics as 'essential features' - after all, a point pusher would crash into a wall, which would be quite catastrophic in real life situations, while we have shown ours to turn and align smoothly. We also think the changes in acceleration/deceleration, orientation, and the changed residence times characteristic to both model and experiment are notable for real life applications. The point we are raising is that the point particle pusher as in Zöttl 2012 [29] is fundamentally different from the finite-size squirmer we are modelling to match our experiments, thus requiring a different combination of singularities and their image systems. We have reworded the statement the reviewer noted to clarify our point.

Phase portraits are indeed a widely used analysis tool to prove robust limit cycles (see references Junot 2019 [11], Mathijssen 2019 [12], Katuri 2018 [18], de Graaf 2016 [37], and Zöttl 2012 [29]). We simply use them to illustrate characteristic features of our specific cycles, e.g. the orientational dynamics. We think they provide an instructive visual illustration of the direct effects of including finite size and the associated hydrodynamic interactions. Therefore we also include the limit cycles similar to [29] for comparison (green, grey).

We do agree that oscillatory upstream rheotaxis has been found in biological systems like in parasites (Uppaluri 2012, ref. [6]), and very recently in E. Coli (Junot 2019 and Mathijssen 2019, refs. [11, 12]), as discussed in the introduction. However, to the best of our knowledge, it has not been demonstrated, and analyzed, before for a priori non-oscillating artificial systems.

We thank the reviewer for bringing Caldag 2019 [23] to our attention, an interesting model of magnetically actuated (mm-sized) swimmers with intrinsic shape asymmetry and helical motion, that is "slightly affected by the [superimposed] channel flow" [23]. The helical motion of these swimmers is therefore not generally a consequence of rheotaxis. In contrast, our model swimmers do not have any intrinsic helicity, such that the oscillation is demonstrably a direct consequence of the rheotactic interaction. We believe that [23] would be appropriately addressed in our literature discussion of swimmers with intrinsic broken symmetries and have accordingly done so (see changes in the introduction and refs [23, 24]. in the revised manuscript; p. 1 line 56).

The reviewer also suggested Costanzo 2012 (now ref. [8]), which consists of numerical simulations of model bacteria in 2-D channels. The study is chiefly designed towards understanding of collective behaviour in a shear flow, so rather different from the scope of our work. We note that in section 3.1 of [8], the authors briefly show the oscillatory rheotactic trajectories of a single model cell as obtained from their numerics. However, these do not include hydrodynamic interactions between the model bacteria and the wall (see p. 2 second column of [8]), which in our experiments significantly affect the rheotactic dynamics.

In the following I have more comments regarding the authors' answer to my initially raised points 1. - 11.

Comment to 1.: Thank you for clarifying this point. The Supplementary Figure S5 shows nicely the robustness of the phase space dynamics. Still the authors did not answer my question how many trajectories they observe for the different flow rates.

Reply: We are sorry we forgot to add the numbers. We approximately have 7 independent data sets for limit cycles for a definite flow rate. This information has now been added to the Supplemental Figure S5's caption. Since each data set requires laborious PIV analysis of the external flow profile (and the behaviour is observably steady and reproducible, anyway, see e.g. Video S3), we could not go for mass experimental statistics.

Comment to 2.: As pointed out already, I agree that the experimental system in the underlying manuscript is very nice and presents valuable new results. I disagree however with the novelty aspect regarding the oscillatory dynamics, which has been seen in other theoretical and experimental works.

Reply:

We hope this point is sufficiently addressed by our textual changes.

Comment to 3.: As stated above, of course I agree that including the hydrodynamic interactions is important when comparing quantitatively to the experiments, and in Ref [37] hydrodynamic interactions have been shown to be important to determine the stability of the oscillatory dynamics. I am just not convinced, that the extended model adds fundamental new features. Yes, the shape of the oscillatory trajectories are different when adding more hydrodynamic interaction terms. This is related to the fact that the extra terms shift the 'critical' flow strength which separates the upstream to the downstream trajectories, and hence the shape of the trajectories, the specific values of $\psi(t)$, and the rheotactic velocity at a specific flow rate, but do not change the essential characteristics of the oscillations.

Reply:

We believe this point is best discussed as part our general reply to the reviewer's initial comments, please see above.

Comment to 4.: I am sorry I missed to include the reference. It is Caldag and Yesilyurt, Journal of Fluids and Structures 90, 164, 2019; where the upstream oscillation of artificial pusher and puller swimmers in cylindrical pipe flow has been investigated, and stable upstream oscillations observed, and experiments and simulations compared. See reference [51] below.

Reply:

We thank the reviewer for the clarification, and have briefly addressed above where this study fits into the state of the art. For additional context: these magnetically-actuated, pusher-type swimmers swim along a helical trajectory because of their shape and actuation mechanism *even in absence of any external flow* (see Fig. 6 of Microfluidics and Nanofluidics 19, 1109, 2015; ref. [24]). The actuation of the flow results in some minor alterations of this inherent helical trajectory, *e.g.* changes in wavelength. In relation to the effects of channel flow on swimmer trajectories, the authors conclude in ref. [23] (Caldag 2019, p. 172-173): "In experiments, pullers were slightly affected by the channel flow *while pushers were not affected as much except at high flow rates (Caldag et al., 2017)*". To summarize, at mild flow rates their pusher-mode trajectories were close to those observed without the flow (p. 174 in [23]), while at high flow rates, these pushers were destabilized and crashed onto the channel walls (p. 175 in [23]).

Comment to 5.: Thank you for the clarification regarding the flow profile measurement.

Reply: We are happy that the reviewer is satisfied.

Comment to 6.: I think it is good that the authors showed more explicitly how the hydrodynamic interactions influence the upstream/downstream transition and the shape of the oscillatory trajectories. While this has not been done systematically, it still gives a good impression how the hydrodynamic interactions are able to influence the upstream swimming condition.

Reply:

We are happy that we were able to convey our argument to the reviewer.

Comment to 7.: See my comments above.

Reply: Please see our replies above.

Comment to 8.: Thank you for this extra discussion. As stated above, I would be cautious naming it a 'new rheotactic behavior'.

Reply:

We hope the reviewer is satisfied with our more precise rewording.

Comment to 9.: Yes I see why it makes sense to model the droplet as a squirmer in a quiescent fluid. However, one could think that external shear flow will induce flow of the surfactants at the droplet surface which influence the droplet swimming orientation, in contrast to phoretic swimmers with an intrinsic orientation.

Reply:

We considered chemical feedback during our experiments, and for that reason did the trail fluorescence analysis. An absolutely exact quantitative model should indeed account for these effects, but that would require an elaborate advection-diffusion model for the chemical and hydrodynamic fields, as e.g. used in Choudhary 2021, ref. [45] and related works by the Michelin group (see the Discussion section, p. 9, l. 530). Moreover, this approach is extremely specific to autophoretic droplets, while our description centres more on model independent features. Within the resolution of our experiments, the trail orientation aligns with the hydrodynamic orientation of the swimmer, which suggests that these phoretic effects are at most second order (cf. MS p. 5, l. 311-316 and discussion above that).

Comment to 10.: Thank you for the clarification.

Reply:

We are happy that the reviewer is satisfied.

Comment to 11.: I agree that the verification of the theoretically proposed stable upstream oscillations, as reported in [37], is a valuable and important aspect of the manuscript. I would just be more cautious talking about novel effects, and about proposing a new theoretical model. For me the 'novel' aspect in the paper is the demonstration that the specific hydrodynamic interactions of the active droplet with the boundaries influence the upstream swimming condition and shape of trajectories, and not the oscillatory upstream phenomenon itself. However, this has not been stated clearly in the manuscript. Also, several described phenomena in the manuscript have been found theoretically in [37], but not referred to in the manuscript.

Reply:

We thank the reviewer for their appreciation and their engagement in re-reviewing our study which has helped us in sharpening the key findings of our study. We would like to add that what is important to us beyond the changes in shape of trajectory is also the differences in orientation and velocity of the microswimmer. The method-of-images approach is indeed established; however, by our specific choice of singularities, and their image systems, we do not simply verify generic features of [29], but illustrate the necessary changes between an idealized, point-sized pusher/puller assumption and finite sized swimmers in real life geometries. We hope these points are sufficiently addressed by our comparison to [29] in the introduction.

Reviewer #2 (Remarks to the Author):

I report here on the resubmitted manuscript by Dey et al. In my initial comments, I had three major concerns: (i) the relationship of the results to previously described systems by the same group, (ii) the generality of the results and (iii) a technical comment on the 2d approximation made. First of all, I am happy to see that the authors addressed these comments and that - regarding my third point - the use of a 3d description of the flow profile improves the comparison between the model and the experiments. Regarding the points (i) and (ii) there is a small inconsistency in claiming generality of the oscillatory dynamics on the one hand and arguing that the formulation used in a previous paper do not relate to the observed behaviour. So I remain a little puzzled by these statements. However, I do not think it is necessary to be too picky about this. In the end, the manuscript is a great combination of

experimental results on an original system, thorough modeling and interesting discussion and I would support publication.

Reply:

We thank the reviewer for their time and appreciation and the valuable improvements they suggested in round 1.

Reviewer #3 (Remarks to the Author):

Dear Editors,

The authors addressed my first comment on the definitions of two angles. However, I don't feel that the answer to question 2 on the choice of the truncated model is satisfactory. It is hard to believe that one needs 12 terms in the expansion for the occurrence of a limit cycle. I am pretty sure that a much simple system can capture a limit cycle. Furthermore, the comparison with the experiment shown in Fig. 5b is somewhat qualitative. Thus, a much simpler minimal model would provide a better insight than the complicated one.

Reply:

The reviewer is right in their remark that one does not need all the 12 image systems to get a generic oscillatory behaviour, and hence, *some* limit cycle. We demonstrate this using the grey and the green oscillatory trajectories in Fig. 5(a) and 5b(ii). The grey trajectory excludes microswimmer-wall hydrodynamic interaction: it is evaluated without using any image term. The green trajectory considers only the image of the force-dipole singularity (the highest order singularity in the squirmer model for a force-free microswimmer). A comparison (Fig. 5b) of the phase space trajectories with our experimental data (markers) and the results of our extended hydrodynamic model (dark red limit cycle) shows that these minimal models predict a different shape of the experimentally observed limit cycle.

Our choice of exactly these 12 images is motivated as follows. There are three singularities. Physically, the source dipole takes care of the finite size of the swimmer, essentially by defining a surface around the swimmer through which no fluid passes; the force dipole addresses the pusher characteristic of the swimming mechanism; and the source quadrupole addresses the hydrodynamics closer to the microswimmer (in the near field region; hence, it influences the trajectory shape when the microswimmer is closer to the wall).

Since the channel has four walls, we include images at all four. One might argue that top and bottom could be omitted since the confinement is here strong enough to be modelled as quasi-2D, but it felt inconsistent to us to ignore the two walls closest to the droplet. While this does result in having to consider 12 instead of 6 image systems, the resulting equations of motion only gain a constant contribution (which makes sense, because nothing oscillates in z). However, this contribution affects the quantitative dynamics of the system (as sort of a viscous resistance from the top and bottom walls due to the finite size of the microswimmer), and should therefore be included in direct experimental comparisons.

The physical motivations for each chosen singularity are summarised in the full derivation of the image model in the Methods section (MS l. 706, p. 11), so they are currently not included of the manuscript itself to avoid repetition.

Importantly, matching the actual shape of the limit cycle provides insights into the specific features of the rheotactic dynamics in a strong confinement that we have demonstrated in the manuscript. This holds not just for academic purposes, but will have consequences for real life applications.

Minimal models differ from finite-size models like ours: they predict bouncing-off or crashing of a microswimmer onto a confinement wall, underestimate the residence time of a microswimmer at interfaces, and predict a different acceleration/deceleration (hence the rheotactic velocity) and orientation of the microswimmer, as demonstrated by our grey and green data sets. A key conclusion of our study is that to capture the rheotactic dynamics of a finite size self-propelling microswimmer in a strong confinement in a physically consistent model, one has to consider the microswimmer-wall hydrodynamic interactions.

Figure 1: Comparison between the experimentally (markers) and theoretically (solid red line) obtained stable limit cycles for a definite imposed flow rate

Regarding the direct comparison of limit cycles in theory and experiment in fig 5b: Our model tries to theoretically capture the finite-size effect of the experimental droplet through the source dipole singularity. The inclusion of the source dipole singularity allows the model squirmer to consistently turn away from the approaching wall by imposing a closest distance of approach, and hence, imposes an effective size. We can therefore expect some mismatch between experiment and theory in regard to the closest distance that the microswimmer can actually swim to relative to the side walls of the microchannel (Fig. 5bi). However, there is good quantitative agreement between the experimental and theoretical variations in microswimmer *orientation*, which is apparent if we replot the phase portrait after rescaling \bar{y} as \bar{y}/\bar{y}_{max} (see Fig. 1).

REVIEWERS' COMMENTS

Reviewer #1 (Remarks to the Author):

I am happy to see that the authors have finally made the necessary amendments in their manuscript clarifying important issues regarding originality and novelty aspects in their work. They have now put their results much more into context of previous work, both theoretical and experimental, and their claims regarding novelty are more precise now. Furthermore, it is more clear now why they have used the specific hydrodynamic model, and what it adds to previous models.

Reviewer #3 (Remarks to the Author):

I am satisfied with the reply and have no further objections.
The manuscript is suitable for publication.

We are very happy all three reviewers are satisfied. They expended an outstanding interest and care in pointing out necessary clarifications and sharpening our key findings, and we truly appreciate their feedback. In view of their respective comments we do not believe a point-by-point response is required.

Reviewer # 1 (Remarks to the Author):

I am happy to see that the authors have finally made the necessary amendments in their manuscript clarifying important issues regarding originality and novelty aspects in their work. They have now put their results much more into context of previous work, both theoretical and experimental, and their claims regarding novelty are more precise now. Furthermore, it is more clear now why they have used the specific hydrodynamic model, and what it adds to previous models.

Reviewer # 3 (Remarks to the Author):

I am satisfied with the reply and have no further objections. The manuscript is suitable for publication.